# DPM-Solver++: Fast Solver for Guided Sampling of Diffusion Probabilistic Models

## Abstract

Diffusion probabilistic models (DPMs) have achieved impressive success in high-resolution image synthesis, especially in recent large-scale text-to-image generation applications. An essential technique for improving the sample quality of DPMs is *guided sampling*, which usually needs a large guidance scale to obtain the best sample quality. The commonly-used fast sampler for guided sampling is DDIM, a first-order diffusion ODE solver that generally needs 100 to 250 steps for high-quality samples. Although recent works propose dedicated high-order solvers and achieve a further speedup for sampling without guidance, their effectiveness for guided sampling has not been well-tested before. In this work, we demonstrate that previous high-order fast samplers suffer from instability issues, and they even become slower than DDIM when the guidance scale grows large. To further speed up guided sampling, we propose *DPM-Solver++*, a high-order solver for the guided sampling of DPMs. DPM-Solver++ solves the diffusion ODE with the data prediction model and adopts thresholding methods to keep the solution matches training data distribution. We further propose a multistep variant of DPM-Solver++ to address the instability issue by reducing the effective step size. Experiments show that DPM-Solver++ can generate high-quality samples within only 15 to 20 steps for guided sampling by pixel-space and latent-space DPMs.

## 1 Introduction

Diffusion probabilistic models (DPMs) (Sohl-Dickstein et al., 2015; Ho et al., 2020; Song et al., 2021b) have achieved impressive success on various tasks, such as high-resolution image synthesis (Dhariwal & Nichol, 2021; Ho et al., 2022; Rombach et al., 2022), image editing (Meng et al., 2022; Saharia et al., 2022a; Zhao et al., 2022), text-to-image generation (Nichol et al., 2021; Saharia et al., 2022b; Ramesh et al., 2022; Rombach et al., 2022; Gu et al., 2022), voice synthesis (Liu et al., 2022a; Chen et al., 2021a;b), molecule generation (Xu et al., 2022; Hoogeboom et al., 2022; Wu et al., 2022) and data compression (Theis et al., 2022; Kingma et al., 2021). Compared with other deep generative models such as GANs (Goodfellow et al., 2014) and VAEs (Kingma & Welling, 2014), DPMs can even achieve better sample quality by leveraging an essential technique called *guided sampling* (Dhariwal & Nichol, 2021; Ho & Salimans, 2021), which uses additional guidance models to improve the sample fidelity and the condition-sample alignment. Through it, DPMs in text-to-image and image-to-image tasks can generate high-resolution photorealistic and artistic images which are highly correlated to the given condition, bringing a new trend in artificial intelligence art painting.

The sampling procedure of DPMs gradually removes the noise from pure Gaussian random variables to obtain clear data, which can be viewed as discretizing either the diffusion SDEs (Ho et al., 2020; Song et al., 2021b) or the diffusion ODEs (Song et al., 2021b;a) defined by a parameterized noise prediction model or data prediction model (Ho et al., 2020; Kingma et al., 2021). Guided sampling of DPMs can also be formalized with such discretizations by combining an unconditional model with a guidance model, where a hyperparameter controls the scale of the guidance model (i.e. *guidance scale*). The commonly-used method for guided sampling is DDIM (Song et al., 2021a), which is proven as a first-order diffusion ODE solver (Salimans & Ho, 2022; Lu et al., 2022) and it generally needs 100 to 250 steps of large neural network evaluations to converge, which is time-consuming.

Dedicated high-order diffusion ODE solvers (Lu et al., 2022; Zhang & Chen, 2022) can generate high-quality samples in 10 to 20 steps for sampling without guidance. However, their effectiveness

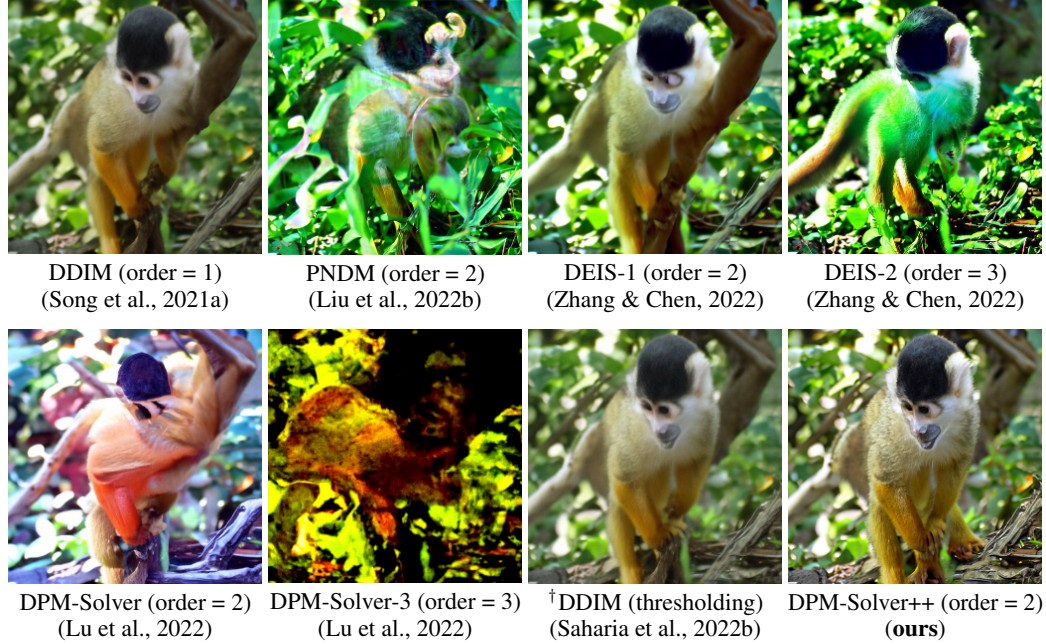

| DDIM (order = 1) | PNDM (order = 2) | DEIS-1 (order = 2) | DEIS-2 (order = 3) |
| (Song et al., 2021a) | (Liu et al., 2022b) | (Zhang & Chen, 2022) | (Zhang & Chen, 2022) |
| DPM-Solver (order = 2) | DPM-Solver-3 (order = 3) | [†]DDIM (thresholding) | DPM-Solver++ (order = 2) |
| (Lu et al., 2022) | (Lu et al., 2022) | (Saharia et al., 2022b) | (**ours**) |

Figure 1: Previous high-order solvers are unstable for guided sampling: Samples using the pre-trained DPMs (Dhariwal & Nichol, 2021) on ImageNet 256×256 with a classifier guidance scale 8.0, varying different samplers (and different solver orders) with only 15 function evaluations. †: DDIM with the dynamic thresholding (Saharia et al., 2022b). Our proposed DPM-Solver++ (detailed in Algorithm 2) can generate better samples than the first-order DDIM, while other high-order samplers are worse than DDIM.

for guided sampling has not been carefully examined before. In this work, we demonstrate that previous high-order solvers for DPMs generate unsatisfactory samples for guided sampling, even worse than the simple first-order solver DDIM. We identify two challenges of applying high-order solvers to guided sampling: (1) the large guidance scale narrows the convergence radius of high-order solvers, making them unstable; and (2) the converged solution does not fall into the same range with the original data (a.k.a. "train-test mismatch" (Saharia et al., 2022b)).

Based on the observations, we propose *DPM-Solver++*, a training-free fast diffusion ODE solver for guided sampling. We find that the parameterization of the DPM critically impacts the solution quality. Subsequently, we solve the diffusion ODE defined by the data prediction model, which predicts the clean data given the noisy ones. We derive a high-order solver for solving the ODE with the data prediction parameterization, and adopt dynamic thresholding methods (Saharia et al., 2022b) to mitigate the train-test mismatch problem. Furthermore, we develop a multistep solver which uses smaller step sizes to address instability.

As shown in Fig. 1, DPM-Solver++ can generate high-quality samples in only 15 steps, which is much faster than all the previous training-free samplers for guided sampling. Our additional experimental results show that DPM-Solver++ can generate high-fidelity samples and almost converge within only 15 to 20 steps, for a wide variety of guided sampling applications, including both pixel-space DPMs and latent-space DPMs.

## 2 DIFFUSION PROBABILISTIC MODELS

In this section, we review diffusion probabilistic models (DPMs) and their sampling methods.

### 2.1 FAST SAMPLING FOR DPMS BY DIFFUSION ODES

Diffusion Probabilistic Models (DPMs) (Sohl-Dickstein et al., 2015; Ho et al., 2020; Song et al., 2021b) gradually add Gaussian noise to a $D$-dimensional random variable $\boldsymbol{x}_0 \in \mathbb{R}^D$ to perturb the corresponding unknown data distribution $q_0(\boldsymbol{x}_0)$ at time $0$ to a simple normal distribution $q_T(\boldsymbol{x}_T) \approx \mathcal{N}(\boldsymbol{x}_T | \boldsymbol{0}, \tilde{\sigma}^2 \boldsymbol{I})$ at time $T > 0$ for some $\tilde{\sigma} > 0$. The transition distribution $q_{t0}(\boldsymbol{x}_t | \boldsymbol{x}_0)$ at

each time $t \in [0, T]$ satisfies

$$q_{t0}(\boldsymbol{x}_t|\boldsymbol{x}_0) = \mathcal{N}(\boldsymbol{x}_t|\alpha_t\boldsymbol{x}_0, \sigma_t^2\boldsymbol{I}), \tag{1}$$

where $\alpha_t, \sigma_t > 0$ and the *signal-to-noise-ratio* (SNR) $\alpha_t^2/\sigma_t^2$ is strictly decreasing w.r.t. $t$ (Kingma et al., 2021). Eq. (1) can be written as $\boldsymbol{x}_t = \alpha_t\boldsymbol{x}_0 + \sigma_t\boldsymbol{\epsilon}$, where $\boldsymbol{\epsilon} \sim \mathcal{N}(\boldsymbol{0}, \boldsymbol{I})$.

**Parameterization: noise prediction and data prediction**  DPMs learn to recover the data $\boldsymbol{x}_0$ based on the noisy input $\boldsymbol{x}_T$ with a sequential denoising procedure. There are two alternative ways to define the model. The *noise prediction model* $\boldsymbol{\epsilon}_\theta(\boldsymbol{x}_t, t)$ attempts to predict the noise $\epsilon$ from the data $\boldsymbol{x}_t$, which optimizes the parameter $\theta$ by the following objective (Ho et al., 2020; Song et al., 2021b):

$$\min_\theta \mathbb{E}_{x_0, \boldsymbol{\epsilon}, t} \left[ \omega(t) \| \boldsymbol{\epsilon}_\theta(\boldsymbol{x}_t, t) - \boldsymbol{\epsilon} \|_2^2 \right], \tag{2}$$

where $\boldsymbol{x}_0 \sim q_0(\boldsymbol{x}_0)$, $\boldsymbol{\epsilon} \sim \mathcal{N}(\boldsymbol{0}, \boldsymbol{I})$, $t \sim \mathcal{U}([0, 1])$, and $\omega(t) > 0$ is a weighting function. Alternatively, the *data prediction model* $\boldsymbol{x}_\theta(\boldsymbol{x}_t, t)$ predicts the original data $\boldsymbol{x}_0$ based on the noisy $\boldsymbol{x}_t$, and its relationship with $\boldsymbol{\epsilon}_\theta(\boldsymbol{x}_t, t)$ is given by $\boldsymbol{x}_\theta(\boldsymbol{x}_t, t) := (\boldsymbol{x}_t - \sigma_t\boldsymbol{\epsilon}_\theta(\boldsymbol{x}_t, t))/\alpha_t$ (Kingma et al., 2021).

**Diffusion ODEs**  Sampling by DPMs can be implemented by solving the *diffusion ODEs* (Song et al., 2021b;a; Liu et al., 2022b; Zhang & Chen, 2022; Lu et al., 2022), which is generally faster than other sampling methods. Specifically, sampling by diffusion ODEs need to discretize the following ODE (Song et al., 2021b) with $t$ changing from $T$ to $0$:

$$\frac{\mathrm{d}\boldsymbol{x}_t}{\mathrm{d}t} = f(t)\boldsymbol{x}_t + \frac{g^2(t)}{2\sigma_t}\boldsymbol{\epsilon}_\theta(\boldsymbol{x}_t, t), \quad \boldsymbol{x}_T \sim \mathcal{N}(\boldsymbol{0}, \tilde{\sigma}^2\boldsymbol{I}), \tag{3}$$

and the equivalent diffusion ODE w.r.t. the data prediction model $\boldsymbol{x}_\theta$ is

$$\frac{\mathrm{d}\boldsymbol{x}_t}{\mathrm{d}t} = \left( f(t) + \frac{g^2(t)}{2\sigma_t^2} \right) \boldsymbol{x}_t - \frac{\alpha_t g^2(t)}{2\sigma_t^2}\boldsymbol{x}_\theta(\boldsymbol{x}_t, t), \quad \boldsymbol{x}_T \sim \mathcal{N}(\boldsymbol{0}, \tilde{\sigma}^2\boldsymbol{I}), \tag{4}$$

where the coefficients $f(t) = \frac{\mathrm{d}\log\alpha_t}{\mathrm{d}t}$, $g^2(t) = \frac{\mathrm{d}\sigma_t^2}{\mathrm{d}t} - 2\frac{\mathrm{d}\log\alpha_t}{\mathrm{d}t}\sigma_t^2$ (Kingma et al., 2021).

## 2.2 Guided Sampling for DPMs

*Guided sampling* (Dhariwal & Nichol, 2021; Ho & Salimans, 2021) is a widely-used technique to apply DPMs for conditional sampling, which is useful in text-to-image, image-to-image, and class-to-image applications (Dhariwal & Nichol, 2021; Saharia et al., 2022b; Rombach et al., 2022; Nichol et al., 2021; Ramesh et al., 2022). Given a condition variable $c$, guided sampling defines a conditional noise prediction model $\tilde{\boldsymbol{\epsilon}}_\theta(\boldsymbol{x}_t, t, c)$. There are two types of guided sampling methods, depending on whether they require a classifier model. *Classifier guidance* (Dhariwal & Nichol, 2021) leverages a pretrained classifier $p_\phi(c|\boldsymbol{x}_t, t)$ to define the conditional noise prediction model by:

$$\tilde{\boldsymbol{\epsilon}}_\theta(\boldsymbol{x}_t, t, c) := \boldsymbol{\epsilon}_\theta(\boldsymbol{x}_t, t) - s \cdot \sigma_t \nabla_{\boldsymbol{x}_t} \log p_\phi(c|\boldsymbol{x}_t, t), \tag{5}$$

where $s > 0$ is the *guidance scale*. In practice, a large $s$ is usually preferred for improving the condition-sample alignment (Rombach et al., 2022; Saharia et al., 2022b) for guided sampling. *Classifier-free guidance* (Ho & Salimans, 2021) shares the same parameterized model $\boldsymbol{\epsilon}_\theta(\boldsymbol{x}_t, t, c)$ for the unconditional and conditional noise prediction models, where the input $c$ for the unconditional model is a special placeholder $\varnothing$. The corresponding conditional model is defined by:

$$\tilde{\boldsymbol{\epsilon}}_\theta(\boldsymbol{x}_t, t, c) := s \cdot \boldsymbol{\epsilon}_\theta(\boldsymbol{x}_t, t, c) + (1 - s) \cdot \boldsymbol{\epsilon}_\theta(\boldsymbol{x}_t, t, \varnothing). \tag{6}$$

Then, samples can be drawn by solving the ODE (3) with $\boldsymbol{\epsilon}_\theta(\boldsymbol{x}_t, t, c)$ in place of $\boldsymbol{\epsilon}_\theta(\boldsymbol{x}_t, t)$. DDIM (Song et al., 2021a) is a typical solver for guided sampling, which generates samples in a few hundreds of steps.

## 2.3 Exponential Integrators and High-Order ODE Solvers

It is shown in recent works (Lu et al., 2022; Zhang & Chen, 2022) that ODE solvers based on exponential integrators (Hochbruck & Ostermann, 2010) converge much faster than the traditional

solvers for solving the unconditional diffusion ODE (3). Given an initial value $\boldsymbol{x}_s$ at time $s > 0$, Lu et al. (2022) derive the solution $\boldsymbol{x}_t$ of the diffusion ODE (3) at time $t$ as:

$$\boldsymbol{x}_t = \frac{\alpha_t}{\alpha_s}\boldsymbol{x}_s - \alpha_t \int_{\lambda_s}^{\lambda_t} e^{-\lambda}\hat{\boldsymbol{\epsilon}}_\theta(\hat{\boldsymbol{x}}_\lambda, \lambda)\mathrm{d}\lambda, \tag{7}$$

where the ODE is changed from the time ($t$) domain to the log-SNR ($\lambda$) domain by the change-of-variables formula. Here, the log-SNR $\lambda_t := \log(\alpha_t/\sigma_t)$ is a strictly decreasing function of $t$ with the inverse function $t_\lambda(\cdot)$, and $\hat{\boldsymbol{x}}_\lambda := \boldsymbol{x}_{t_\lambda(\lambda)}$, $\hat{\boldsymbol{\epsilon}}_\theta(\hat{\boldsymbol{x}}_\lambda, \lambda) := \boldsymbol{\epsilon}_\theta(\boldsymbol{x}_{t_\lambda(\lambda)}, t_\lambda(\lambda))$ are the corresponding change-of-variable forms for $\lambda$. Lu et al. (2022) showed that DDIM is a first-order solver for Eq. (7). They further proposed a high-order solver named "DPM-Solver", which can generate realistic samples for the unconditional model in only 10-20 steps.

Unfortunately, the outstanding efficiency of existing high-order solvers does not transfer to guided sampling, which we shall discuss soon.

## 3 CHALLENGES OF HIGH-ORDER SOLVERS FOR GUIDED SAMPLING

Before developing new fast solvers, we first examine the performance of existing high-order diffusion ODE solvers and highlight the challenges.

The first challenge is *the large guidance scale causes high-order solvers to be instable*. As shown in Fig. 1, for a large guidance scale $s = 8.0$ and 15 function evaluations, previous high-order diffusion ODE solvers (Lu et al., 2022; Zhang & Chen, 2022; Liu et al., 2022b) produce low-quality images. Their sample quality is even worse than the first-order DDIM. Moreover, the sample quality becomes even worse as the order of the solver gets higher.

Intuitively, large guidance scales may amplify both the output and the derivatives of the model $\tilde{\boldsymbol{\epsilon}}_\theta$ in Eq. (5). The derivatives of the model affect the convergence range of ODE solvers, and the amplification may cause high-order ODE solvers to need much smaller step sizes to converge, and thus the higher-order solvers may perform worse than the first-order solver. Moreover, high-order solvers require high-order derivatives, which are generally more sensitive to the amplifications. This further narrows the convergence radius.

The second challenge is *the "train-test mismatch" problem* (Saharia et al., 2022b). The data lie in a bounded interval (e.g. $[-1, 1]$ for image data). However, the large guidance scale pushes the conditional noise prediction model $\tilde{\boldsymbol{\epsilon}}_\theta(\boldsymbol{x}_t, t, c)$ away from the true noise, which in turns make the sample (i.e. the converged solution $\boldsymbol{x}_0$ of diffusion ODEs) to fall out of the bound. In this case, the generated images are saturated and unnatural (Saharia et al., 2022b).

## 4 DESIGNING TRAINING-FREE FAST SAMPLERS FOR GUIDED SAMPLING

In this section, we design novel high-order diffusion ODE solvers for faster guided sampling. As discussed in Sec. 3, previous high-order solvers have instability and "train-test mismatch" issues for large guidance scales. The "train-test mismatch" issue arises from the ODE itself, and we find the parameterization of the ODE is critical for the converged solution to be bounded. While previous high-order solvers are designed for the noise prediction model $\tilde{\boldsymbol{\epsilon}}_\theta$, we solve the ODE (4) for the data prediction model $\boldsymbol{x}_\theta$, which itself has some advantages and thresholding methods are further available to keep the samples bounded (Ho et al., 2020; Saharia et al., 2022b). We also propose a multistep solver to address the instability issue.

### 4.1 DESIGNING SOLVERS BY DATA PREDICTION MODEL

We follow the notations in Lu et al. (2022). Given a sequence $\{t_i\}_{i=0}^M$ decreasing from $t_0 = T$ to $t_M = 0$ and an initial value $\boldsymbol{x}_{t_0} \sim \mathcal{N}(\boldsymbol{0}|\tilde{\sigma}^2\boldsymbol{I})$, the solver aims to iteratively compute a sequence $\{\tilde{\boldsymbol{x}}_{t_i}\}_{i=0}^M$ to approximate the exact solution at each time $t_i$, and the final value $\tilde{\boldsymbol{x}}_{t_M}$ is the approximated sample by the diffusion ODE. Denote $h_i := \lambda_{t_i} - \lambda_{t_{i-1}}$ for $i = 1, \ldots, M$.

For solving the diffusion ODE w.r.t. $\boldsymbol{x}_\theta$ in Eq. (4), we firstly propose a simplified formulation of the exact solution of diffusion ODEs w.r.t. $\boldsymbol{x}_\theta$ below. Such formulation exactly computes

the linear term in Eq. (4) and only remains an exponentially-weighted integral of $\boldsymbol{x}_\theta$. Denote $\hat{\boldsymbol{x}}_\theta(\hat{\boldsymbol{x}}_\lambda, \lambda) \coloneqq \boldsymbol{x}_\theta(\boldsymbol{x}_{t_\lambda(\lambda)}, t_\lambda(\lambda))$ as the change-of-variable form of $\boldsymbol{x}_\theta$ for $\lambda$, we have:

**Proposition 4.1** (Exact solution of diffusion ODEs of $\boldsymbol{x}_\theta$, proof in Appendix A)**.** *Given an initial value $\boldsymbol{x}_s$ at time $s > 0$, the solution $\boldsymbol{x}_t$ at time $t \in [0, s]$ of diffusion ODEs in Eq. (4) is:*

$$\boldsymbol{x}_t = \frac{\sigma_t}{\sigma_s}\boldsymbol{x}_s + \sigma_t \int_{\lambda_s}^{\lambda_t} e^\lambda \hat{\boldsymbol{x}}_\theta(\hat{\boldsymbol{x}}_\lambda, \lambda) \mathrm{d}\lambda. \tag{8}$$

As the diffusion ODEs in Eq. (3) and Eq.(4) are equivalent, the exact solution formulations in Eq. (7) and Eq. (8) are also equivalent. However, from the prespective of designing ODE solvers, these two formulations are different. Firstly, Eq. (7) exactly computes the linear term $\frac{\alpha_t}{\alpha_s}\boldsymbol{x}_s$, while Eq. (8) exactly computes another the linear term $\frac{\sigma_t}{\sigma_s}\boldsymbol{x}_s$. Moreover, to design ODE solvers, Eq. (7) needs to approximate the integral $\int e^{-\lambda}\boldsymbol{\epsilon}_\theta \mathrm{d}\lambda$, while Eq. (8) needs to approximate $\int e^\lambda \boldsymbol{x}_\theta \mathrm{d}\lambda$, and these two integrals are different (recall that $\boldsymbol{x}_\theta \coloneqq (\boldsymbol{x}_t - \sigma_t \boldsymbol{\epsilon}_\theta)/\alpha_t$). Therefore, the high-order solvers based on Eq. (7) and Eq. (8) are essentially different. We further propose the general manner for design high-order ODE solvers based on Eq. (8) below.

Given the previous value $\tilde{\boldsymbol{x}}_{t_{i-1}}$ at time $t_{i-1}$, the aim of our solver is to approximate the exact solution at time $t_i$. Denote $\boldsymbol{x}_\theta^{(n)}(\lambda) \coloneqq \frac{\mathrm{d}^n \hat{\boldsymbol{x}}_\theta(\boldsymbol{x}_\lambda, \lambda)}{\mathrm{d}\lambda^n}$ as the $n$-th order total derivatives of $\boldsymbol{x}_\theta$ w.r.t. logSNR $\lambda$. For $k \geq 1$, taking the $(k-1)$-th Taylor expansion at $\lambda_{t_{i-1}}$ for $\boldsymbol{x}_\theta$ w.r.t. $\lambda \in [\lambda_{t_{i-1}}, \lambda_{t_i}]$ and substitute it into Eq. (8) with $s = t_{i-1}$ and $t = t_i$, we have

$$\tilde{\boldsymbol{x}}_{t_i} = \frac{\sigma_{t_i}}{\sigma_{t_{i-1}}}\tilde{\boldsymbol{x}}_{t_{i-1}} + \sigma_{t_i} \sum_{n=0}^{k-1} \underbrace{\boldsymbol{x}_\theta^{(n)}(\hat{\boldsymbol{x}}_{\lambda_{t_{i-1}}}, \lambda_{t_{i-1}})}_{\text{estimated}} \underbrace{\int_{\lambda_{t_{i-1}}}^{\lambda_{t_i}} e^\lambda \frac{(\lambda - \lambda_{t_{i-1}})^n}{n!}\mathrm{d}\lambda}_{\text{analytically computed (Appendix A)}} + \underbrace{\mathcal{O}(h_i^{k+1})}_{\text{omitted}}, \tag{9}$$

where the integral $\int e^\lambda \frac{(\lambda - \lambda_{t_{i-1}})^n}{n!}\mathrm{d}\lambda$ can be **analytically** computed by integral-by-parts (detailed in Appendix A). Therefore, to design a $k$-th order ODE solver, we only need to estimate the $n$-th order derivatives $\boldsymbol{x}_\theta^{(n)}(\lambda_{t_{i-1}})$ for $n \leq k-1$ after omitting the $\mathcal{O}(h_i^{k+1})$ high-order error terms, which are well-studied techniques and we discussed in details in Sec. 4.2. A special case is $k = 1$, where the solver is the same as DDIM (Song et al., 2021a), and we discuss in Sec. 5.1.

For $k = 2$, we use a similar technique as DPM-Solver-2 (Lu et al., 2022) to estimate the derivative $\boldsymbol{x}_\theta^{(1)}(\hat{\boldsymbol{x}}_{\lambda_{t_{i-1}}}, \lambda_{t_{i-1}})$. Specifically, we introduce an additional intermediate time step $s_i$ between $t_{i-1}$ and $t_i$ and combine the function values at $s_i$ and $t_{i-1}$ to approximate the derivative, which is the standard manner for *singlestep* ODE solvers (Atkinson et al., 2011). Overall, we need $2M + 1$ time steps ($\{t_i\}_{i=0}^M$ and $\{s_i\}_{i=1}^M$) which satisfies $t_0 > s_1 > t_1 > \cdots > t_{M-1} > s_M > t_M$. The detailed algorithm is proposed in Algorithm 1, where we combine the previous value $\tilde{\boldsymbol{x}}_{t_{i-1}}$ at time $t_{i-1}$ with the intermediate value $\boldsymbol{u}_i$ at time $s_i$ to compute the value $\tilde{\boldsymbol{x}}_{t_i}$ at time $t_i$.

---

**Algorithm 1** DPM-Solver++(2S).

**Require:** initial value $\boldsymbol{x}_T$, time steps $\{t_i\}_{i=0}^M$ and $\{s_i\}_{i=1}^M$, data prediction model $\boldsymbol{x}_\theta$.
1: $\tilde{\boldsymbol{x}}_{t_0} \leftarrow \boldsymbol{x}_T$.
2: **for** $i \leftarrow 1$ to $M$ **do**
3:    $h_i \leftarrow \lambda_{t_i} - \lambda_{t_{i-1}}$
4:    $r_i \leftarrow \frac{\lambda_{s_i} - \lambda_{t_{i-1}}}{h_i}$
5:    $\boldsymbol{u}_i \leftarrow \frac{\sigma_{s_i}}{\sigma_{t_{i-1}}}\tilde{\boldsymbol{x}}_{t_{i-1}} - \alpha_{s_i}\left(e^{-r_i h_i} - 1\right)\boldsymbol{x}_\theta(\tilde{\boldsymbol{x}}_{t_{i-1}}, t_{i-1})$
6:    $\boldsymbol{D}_i \leftarrow (1 - \frac{1}{2r_i})\boldsymbol{x}_\theta(\tilde{\boldsymbol{x}}_{t_{i-1}}, t_{i-1}) + \frac{1}{2r_i}\boldsymbol{x}_\theta(\boldsymbol{u}_i, s_i)$
7:    $\tilde{\boldsymbol{x}}_{t_i} \leftarrow \frac{\sigma_{t_i}}{\sigma_{t_{i-1}}}\tilde{\boldsymbol{x}}_{t_{i-1}} - \alpha_{t_i}\left(e^{-h_i} - 1\right)\boldsymbol{D}_i$
8: **end for**
9: **return** $\tilde{\boldsymbol{x}}_{t_M}$

---

**Algorithm 2** DPM-Solver++(2M).

**Require:** initial value $\boldsymbol{x}_T$, time steps $\{t_i\}_{i=0}^M$, data prediction model $\boldsymbol{x}_\theta$.
1: Denote $h_i \coloneqq \lambda_{t_i} - \lambda_{t_{i-1}}$ for $i = 1, \ldots, M$.
2: $\tilde{\boldsymbol{x}}_{t_0} \leftarrow \boldsymbol{x}_T$. Initialize an empty buffer $Q$.
3: $Q \xleftarrow{\text{buffer}} \boldsymbol{x}_\theta(\tilde{\boldsymbol{x}}_{t_0}, t_0)$
4: $\tilde{\boldsymbol{x}}_{t_1} \leftarrow \frac{\sigma_{t_1}}{\sigma_{t_0}}\tilde{\boldsymbol{x}}_0 - \alpha_{t_1}\left(e^{-h_1} - 1\right)\boldsymbol{x}_\theta(\tilde{\boldsymbol{x}}_{t_0}, t_0)$
5: $Q \xleftarrow{\text{buffer}} \boldsymbol{x}_\theta(\tilde{\boldsymbol{x}}_{t_1}, t_1)$
6: **for** $i \leftarrow 2$ to $M$ **do**
7:    $r_i \leftarrow \frac{h_{i-1}}{h_i}$
8:    $\boldsymbol{D}_i \leftarrow \left(1 + \frac{1}{2r_i}\right)\boldsymbol{x}_\theta(\tilde{\boldsymbol{x}}_{t_{i-1}}, t_{i-1}) - \frac{1}{2r_i}\boldsymbol{x}_\theta(\tilde{\boldsymbol{x}}_{t_{i-2}}, t_{i-2})$
9:    $\tilde{\boldsymbol{x}}_{t_i} \leftarrow \frac{\sigma_{t_i}}{\sigma_{t_{i-1}}}\tilde{\boldsymbol{x}}_{t_{i-1}} - \alpha_{t_i}\left(e^{-h_i} - 1\right)\boldsymbol{D}_i$
10:    If $i < M$, then $Q \xleftarrow{\text{buffer}} \boldsymbol{x}_\theta(\tilde{\boldsymbol{x}}_{t_i}, t_i)$
11: **end for**
12: **return** $\tilde{\boldsymbol{x}}_{t_M}$

---

We name the algorithm as DPM-Solver++(2S), which means that the proposed solver is a second-order singlestep method. We present the theoretical guarantee of the convergence order in Appendix A. For

Table 1: Comparison between high-order diffusion ODE solvers based on exponential integrators, including DEIS (Zhang & Chen, 2022), DPM-Solver (Lu et al., 2022) and DPM-Solver++ (ours).

|  | DEIS (Zhang & Chen, 2022) | DPM-Solver (Lu et al., 2022) | DPM-Solver++ (**ours**) |
| --- | --- | --- | --- |
| First-Order | DDIM | DDIM | DDIM |
| Model Type | $\epsilon_\theta$ | $\epsilon_\theta$ | $x_\theta$ |
| Taylor Expansion | $\epsilon_\theta$ for $t$ | $\hat{\epsilon}_\theta$ for $\lambda$ | $\hat{x}_\theta$ for $\lambda$ |
| Solver Type (High-Order) | Multistep | Singlestep | Singlestep + Multistep |

$k \geq 3$, as discussed in Sec. 3, high-order solvers may be unsuitable for large guidance scales, thus we mainly consider $k = 2$ in this work, and leave the solvers for higher orders for future study.

Moreover, we provide a theoretical comparison between DPM-Solver-2 (Lu et al., 2022) and DPM-Solver++(2S) in Appendix B. We find that DPM-Solver++(2S) has a smaller constant before the high-order error terms, thus generally has a smaller discretization error than DPM-Solver-2.

### 4.2 From Singlestep to Multistep

At each step (from $t_{i-1}$ to $t_i$), the proposed singlestep solver needs two sequential function evaluations of the neural network $x_\theta$. Moreover, the intermediate values $u_i$ are only used once and then discarded. Such method loses the previous information and may be inefficient. In this section, we propose another second-order diffusion ODE solver which uses the previous information at each step.

In general, to approximate the derivatives $x_\theta^{(n)}$ in Eq. (9) for $n \geq 1$, there is another mainstream approach (Atkinson et al., 2011): *multistep* methods (such as Adams–Bashforth methods). Given the previous values $\{\tilde{x}_{t_j}\}_{j=0}^{i-1}$ at time $t_{i-1}$, multistep methods just reuse the previous values to approximate the high-order derivatives. Multistep methods are empirically more efficient than singlestep methods, especially for limited number of function evaluations. (Atkinson et al., 2011)

We combine the techniques for designing multistep solvers with the Taylor expansions in Eq. (9) and further propose a multistep second-order solver for diffusion ODEs with $x_\theta$. The detailed algorithm is proposed in Algorithm 2, where we combine the previous values $\tilde{x}_{t_{i-1}}$ and $\tilde{x}_{t_{i-2}}$ to compute the value $\tilde{x}_{t_i}$ without additional intermediate values. We name the algorithm as DPM-Solver++(2M), which means that the proposed solver is a second-order multistep solver. We also present a detailed theoretical guarantee of the convergence order, which is stated in Appendix A.

For a fixed budget $N$ of the total number of function evaluations, multistep methods can use $M = N$ steps, while $k$-th order singlestep methods can only use no more than $M = N/k$ steps. Therefore, each step size $h_i$ of multistep methods is around $1/k$ of that of singlestep methods, so the high-order error terms $\mathcal{O}(h_i^k)$ in Eq. (9) of multistep methods may also be smaller than those of singlestep methods. We show in Sec. 6.1 that the multistep methods are slightly better than singlestep methods.

### 4.3 Combining Thresholding with DPM-Solver++

For distributions of bounded data (such as the image data), thresholding methods (Ho et al., 2020; Saharia et al., 2022b) can push out-of-bound samples inwards and somehow reduce the adverse impact of the large guidance scale. Specifically, thresholding methods define a clipped data prediction model $\hat{x}_\theta(x_t, t, c)$ by elementwise clipping the original model $x_\theta := (x_t - \sigma_t \epsilon_\theta)/\alpha_t$ within the data bound, which results in better sample quality for large guidance scales (Saharia et al., 2022b). As our proposed DPM-Solver++ is designed for the $x_\theta$ model, we can straightforwardly combine thresholding methods with DPM-Solver++.

## 5 Relationship with Other Fast Sampling Methods

In essence, all training-free sampling methods for DPMs can be understood as either discretizing *diffusion SDEs* (Ho et al., 2020; Song et al., 2021b; Jolicoeur-Martineau et al., 2021; Tachibana et al., 2021; Kong & Ping, 2021; Bao et al., 2022b; Zhang et al., 2022) or discretizing *diffusion ODEs* (Song

et al., 2021b;a; Liu et al., 2022b; Zhang & Chen, 2022; Lu et al., 2022). As DPM-Solver++ is designed for solving diffusion ODEs, in this section, we discuss the relationship between DPM-Solver++ and other diffusion ODE solvers. We further briefly discuss other fast sampling methods for DPMs.

## 5.1 COMPARISON WITH SOLVERS BASED ON EXPONENTIAL INTEGRATORS

Previous state-of-the-art fast diffusion ODE solvers (Lu et al., 2022; Zhang & Chen, 2022) leverages exponential integrators to solve diffusion ODEs with noise prediction models $\epsilon_\theta$. In short, these solvers approximate the exact solution in Eq. (7) and include DDIM (Song et al., 2021a) as the first-order case. Below we show that the first-order case for DPM-Solver++ is also DDIM.

For $k = 1$, Eq. (9) becomes (after omitting the $\mathcal{O}(h_i^{k+1})$ terms)

$$\tilde{x}_{t_i} = \frac{\sigma_{t_i}}{\sigma_{t_{i-1}}}\tilde{x}_{t_{i-1}} + \sigma_{t_i}x_\theta(\tilde{x}_{t_{i-1}}, t_{i-1})\int_{\lambda_{t_{i-1}}}^{\lambda_{t_i}} e^\lambda \mathrm{d}\lambda$$
$$= \frac{\sigma_{t_i}}{\sigma_{t_{i-1}}}\tilde{x}_{t_{i-1}} - \alpha_{t_i}(e^{-h_i} - 1)x_\theta(\tilde{x}_{t_{i-1}}, t_{i-1}),$$

Therefore, our proposed DPM-Solver++ is the high-order generalization of DDIM w.r.t. the data prediction model $x_\theta$. To the best of our knowledge, such generalization has not been proposed before. We list the detailed difference between previous high-order solvers based on exponential integrators and DPM-Solver++ in Table 1. We emphasize that although the first-order version of these solvers are equivalent, the high-order versions of these solvers are rather different.

## 5.2 OTHER FAST SAMPLING METHODS

Samplers based on diffusion SDEs (Ho et al., 2020; Song et al., 2021b; Jolicoeur-Martineau et al., 2021; Tachibana et al., 2021; Kong & Ping, 2021; Bao et al., 2022b; Zhang et al., 2022) generally needs more steps to converge than those based on diffusion ODEs (Lu et al., 2022), because SDEs introduce more randomness and make denoising more difficult. Samplers based on extra training include model distillation (Salimans & Ho, 2022; Luhman & Luhman, 2021), learning reverse process variances (San-Roman et al., 2021; Nichol & Dhariwal, 2021; Bao et al., 2022a), and learning sampling steps (Lam et al., 2021; Watson et al., 2022). However, training-based samplers are hard to scale-up to pre-trained large DPMs (Saharia et al., 2022b; Rombach et al., 2022; Ramesh et al., 2022). There are other fast sampling methods by modifying the original DPMs to a latent space (Vahdat et al., 2021) or with momentum (Dockhorn et al., 2022). In addition, combining DPMs with GANs (Xiao et al., 2022; Wang et al., 2022) improves the sample quality of GANs and sampling speed of DPMs.

## 6 EXPERIMENTS

In this section, we show that DPM-Solver++ can speed up both the pixel-space DPMs and the latent-space DPMs for guided sampling. We vary different number of function evaluations (NFE) which is the numebr of calls to the model $\epsilon_\theta(x_t, t, c)$ or $x_\theta(x_t, t, c)$, and compare DPM-Solver++ with the previous state-of-the-art fast samplers for DPMs including DPM-Solver (Lu et al., 2022), DEIS (Zhang & Chen, 2022), PNDM (Liu et al., 2022b) and DDIM (Song et al., 2021a). We also convert the discrete-time DPMs to the continuous-time and use these continuous-time solvers. We refer to Appendix C for the detailed implementations and experiment settings.

As previous solvers did not test the performance in guided sampling, we also carefully tune the baseline samplers by ablating the step size schedule (i.e. the choice for the time steps $\{t_i\}_{i=0}^M$) and the solver order. We find that

- For the step size schedule, we search the time steps in the following choices: uniform $t$ (a widely-used setting in high-resolution image synthesis), uniform $\lambda$ (used in (Lu et al., 2022)), uniform split of the power functions of $t$ (used in (Zhang & Chen, 2022), detailed in Appendix C), and we find that the best choice is uniform $t$. Thus, we use uniform $t$ for the time steps in all of our experiments for all of the solvers.
- We find that for a large guidance scale, the best choice for all the previous solvers is the second-order (i.e. DPM-Solver-2 and DEIS-1). However, for a comprehensive comparison,

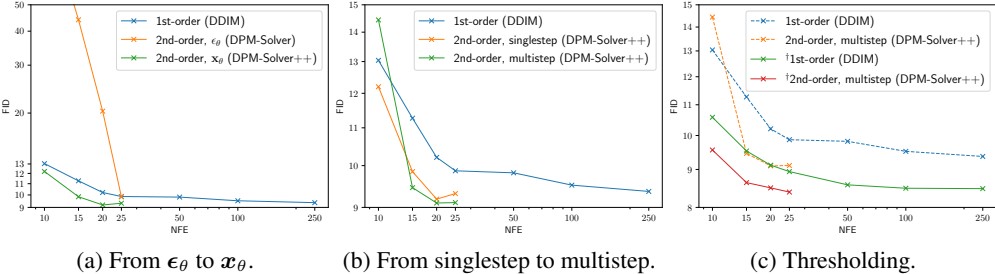

(a) From $\epsilon_\theta$ to $\boldsymbol{x}_\theta$.      (b) From singlestep to multistep.      (c) Thresholding.

Figure 2: Ablation study for DPM-Solver++. Sample quality measured by FID ↓ of different sampling methods for DPMs on ImageNet 256x256 with guidance scale 8.0, varying the number of function evaluations (NFE).

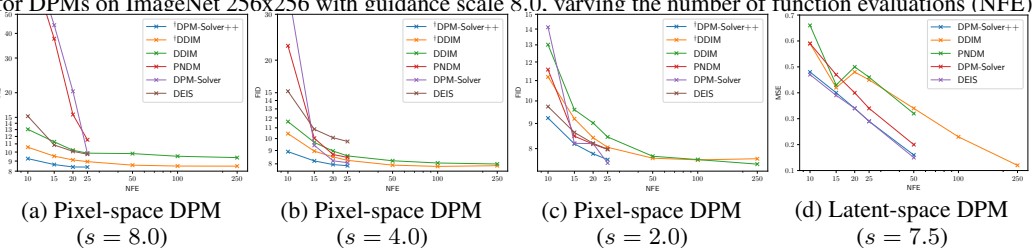

(a) Pixel-space DPM    (b) Pixel-space DPM    (c) Pixel-space DPM    (d) Latent-space DPM
      ($s = 8.0$)           ($s = 4.0$)           ($s = 2.0$)           ($s = 7.5$)

Figure 3: (a-c) Sample quality measured by FID ↓ of different sampling methods for DPMs on ImageNet 256x256 with different guidance scale $s$, varying the number of function evaluations (NFE). †: results by combining the solver with dynamic thresholding method (Saharia et al., 2022b). (d) Convergence error measured by L2-norm ↓ (divided by dimension) between different sampling methods and 1000-step DDIM, varying the number of function evaluations (NFE), for the latent-space DPM "stable-diffusion" (Rombach et al., 2022) on MS-COCO2014 validation set, with the default guidance scale $s = 7.5$ in their official code.

> we run all the orders of previous solvers, including DPM-Solver-2 and DPM-Solver-3; DEIS-1, DEIS-2 and DEIS-3 and choose their best result for each NFE in our comparison.

We run both DPM-Solver++(2S) and DPM-Solver++(2M), and we find that for large guidance scales, the multistep DPM-Solver++(2M) performs better; and for a slightly small guidance scales, the singlestep DPM-Solver++(2S) performs better. We report the best results of DPM-Solver++ and all of the previous samplers in the following sections, the detailed values are listed in Appendix D.

## 6.1 PIXEL-SPACE DPMS WITH GUIDANCE

We firstly compare DPM-Solver++ with other samplers for the guided sampling with classifier-guidance on ImageNet 256x256 dataset by the pretrained DPMs(Dhariwal & Nichol, 2021). We measure the sample quality by drawing 10K samples and computing the widely-used FID score (Heusel et al., 2017), where lower FID usually implies better sample quality. We also adopt the dynamic thresholding method (Saharia et al., 2022b) for both DDIM and DPM-Solver++. We vary the guidance scale $s$ in 8.0, 4.0 and 2.0, the results are shown in Fig. 3(a-c). We find that for large guidance scales, all the previous high-order samplers (DEIS, PNDM, DPM-Solver) converge slower than the first-order DDIM, which shows that previous high-order samplers are unstable. Instead, DPM-Solver++ achieve the best speedup performance for both large guidance scales and small guidance scales. Especially for large guidance scales, DPM-Solver++ can almost converge within only 15 NFE.

As an ablation, we also compare the singlestep DPM-Solver-2, the singlestep DPM-Solver++(2S) and the multistep DPM-Solver++(2M) to demonstrate the effectiveness of our method. We use a large guidance scale $s = 8.0$ and conduct the following ablations:

- From $\epsilon_\theta$ to $\boldsymbol{x}_\theta$: As shown in Fig. 2a, by simply changing the solver from $\epsilon_\theta$ to $\boldsymbol{x}_\theta$ (i.e. from DPM-Solver-2 to DPM-Solver++(2S)), the solver can achieve a stable acceleration performance which is faster than the first-order DDIM. Such result indicates that for guided sampling, high-order solvers w.r.t. $\boldsymbol{x}_\theta$ may be more preferred than those w.r.t. $\epsilon_\theta$.

- From singlestep to multistep: As show in Fig. 2b, the multistep DPM-Solver++(2M) converges slightly faster than the singlestep DPM-Solver++(2S), which almost converges in

15 NFE. Such result indicates that for guided sampling with a large guidance scale, multistep methods may be faster than singlestep methods.

- With or without thresholding: We compare the performance of DDIM and DPM-Solver++ with / without thresholding methods in Fig. 2c. Note that the thresholding method changes the model $x_\theta$ and thus also changes the converged solutions of diffusion ODEs. Firstly, we find that after using the thresholding method, the diffusion ODE can generate higher quality samples, which is consistent with the conclusion in (Saharia et al., 2022b). Secondly, the sample quality of DPM-Solver++ with thresholding outperforms DPM-Solver++ without thresholding under the same NFE. Moreover, when combined with thresholding, DPM-Solver++ is faster than the first-order DDIM, which shows that DPM-Solver++ can also speed up guided sampling by DPMs with thresholding methods.

## 6.2 LATENT-SPACE DPMs WITH GUIDANCE

We also evaluate DPM-Solver++ on the latent-space DPMs (Rombach et al., 2022), which is recently popular among the community due to their official code "stable-diffusion". We use the default guidance scale $s = 7.5$ in their official code. The latent-space DPMs map the image data with a latent code by training a pair of encoder and decoder, and then train a DPM for the latent code. As the latent code is unbounded, we do not apply the thresholding method.

Specifically, we randomly sample 10K caption-image pairs from the MS-COCO2014 validation dataset and use the captions as conditions to draw 10K images from the pretrained "stable-diffusion" model, and we only draw a single image sample of each caption, following the standard evaluation procedures in (Nichol et al., 2021; Rombach et al., 2022). We find that all the solvers can achieve a FID around 15.0 to 16.0 even within only 10 steps, which is very close to the FID computed by the converged samples reported in the official page of "stable-diffusion". We believe it is due to the powerful pretrained decoder, which can map a non-converged latent code to a good image sample.

For latent-space DPMs, different diffusion ODE solvers directly affect the convergence speed on the latent space. To further compare different samplers for latent-space DPMs, we directly compare different solvers according to the convergence error on the latent space by the L2-norm between the sampled $x_0$ and the true solution $x_0^*$ (and the error between them is $\|x_0 - x_0^*\|_2 / \sqrt{D}$). Specifically, we firstly sample 10K noise variables from the standard normal distribution and fix them. Then we sample 10K latent codes by different DPM samplers, starting from the 10K fixed noise variables. As all these solvers can be understood as discretizing diffusion ODEs, we compare the sampled latent codes by the true solution $x_0^*$ from a 999-step DDIM with samples $x_0$ by different samplers within different NFE, and the results are shown in Fig. 3(d). We find that the supported fast samplers (DDIM and PNDM) in "stable-diffusion" converge much slower than DPM-Solver++ and DEIS, and we find that the second-order multistep DPM-Solver++ and DEIS achieve a quite close speedup on the latent space. Moreover, as "stable-diffusion" by default use PNDM with 50 steps, we find that DPM-Solver++ can achieve a similar convergence error with only 15 to 20 steps. We also present an empirical comparison of the sampled images between different solvers in Appendix D, and we find that DPM-Solver++ can indeed generate quite good image samples within only 15 to 20 steps.

## 7 CONCLUSIONS

We study the problem of accelerating guided sampling of DPMs. We demonstrate that previous high-order solvers based on the noise prediction models are abnormally unstable and generate worse-quality samples than the first-order solver DDIM for guided sampling with large guidance scales. To address this issue and speed up guided sampling, we propose DPM-Solver++, a training-free fast diffusion ODE solver for guided sampling. DPM-Solver++ is based on the diffusion ODE with the data prediction models, which can directly adopt the thresholding methods to stabilize the sampling procedure further. We propose both singlestep and multistep variants of DPM-Solver++. Experiment results show that DPM-Solver++ can generate high-fidelity samples and almost converge within only 15 to 20 steps, applicable for pixel-space and latent-space DPMs.

ETHICS STATEMENT

Like other deep generative models such as GANs, DPMs may also be used to generate adverse fake contents (images). The proposed solver can accelerate the guided sampling by DPMs which can further be used for image editing and generate photorealistic fake images. Such influence may further amplify the potential undesirable affects of DPMs for malicious applications.

REPRODUCIBILITY STATEMENT

Our code is based on the official code of DPM-Solver (Lu et al., 2022) and the pretrained checkpoints in Dhariwal & Nichol (2021) and Stable-Diffusion (Rombach et al., 2022). We will release it after the blind review. In addition, datasets used in experiments are publicly available. Our detailed experiment settings and implementations are listed in Appendix C, and the proof of the solver convergence guarantee are presented in Appendix A.

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

# A  ADDITIONAL PROOFS

## A.1  PROOF OF PROPOSITION 4.1

Taking derivative w.r.t. $t$ in Eq. (8) yields

$$
\begin{aligned}
\frac{\mathrm{d}\boldsymbol{x}_t}{\mathrm{d}t} &= \frac{\mathrm{d}\sigma_t}{\mathrm{d}t}\frac{\boldsymbol{x}_s}{\sigma_s} + \frac{\mathrm{d}\sigma_t}{\mathrm{d}t}\int_{\lambda_s}^{\lambda_t} e^\lambda \hat{\boldsymbol{x}}_\theta(\hat{\boldsymbol{x}}_\lambda, \lambda)\mathrm{d}\lambda + \frac{\mathrm{d}\lambda_t}{\mathrm{d}t}\sigma_t e^{\lambda_t}\hat{\boldsymbol{x}}_\theta(\hat{\boldsymbol{x}}_{\lambda_t}, \lambda_t) \\
&= \frac{\mathrm{d}\sigma_t}{\mathrm{d}t}\frac{\boldsymbol{x}_t}{\sigma_t} + \frac{\mathrm{d}\lambda_t}{\mathrm{d}t}\sigma_t e^{\lambda_t}\hat{\boldsymbol{x}}_\theta(\hat{\boldsymbol{x}}_{\lambda_t}, \lambda_t) \\
&= \left(f(t) + \frac{g^2(t)}{2\sigma_t^2}\right)\frac{\boldsymbol{x}_t}{\sigma_t} - \frac{\alpha_t g^2(t)}{2\sigma_t^2}\boldsymbol{x}_\theta(\boldsymbol{x}_t, t),
\end{aligned}
$$

where the last inequality follows from the definitions $f(t) = \frac{\mathrm{d}\log\alpha_t}{\mathrm{d}t}$, $g^2(t) = \frac{\mathrm{d}\sigma_t^2}{\mathrm{d}t} - 2\frac{\mathrm{d}\log\alpha_t}{\mathrm{d}t}\sigma_t^2$.

## A.2  CONVERGENCE OF ALGORITHMS

We make the following assumptions as in Lu et al. (2022) for $\boldsymbol{x}_\theta$, i.e.,

- $\boldsymbol{x}_\theta^{(0)}$, $\boldsymbol{x}_\theta^{(1)}$ and $\boldsymbol{x}_\theta^{(2)}$ exist and are continuous (and hence are bounded).
- The map $\boldsymbol{x} \mapsto \boldsymbol{x}_\theta(\boldsymbol{x}, t)$ is $L$-Lipschitz.
- $h_{max} := \max_{1 \le j \le M} h_j = O(1/M)$.

We also assume further

- $r_i > c > 0$ for all $i = 1, \cdots, M$.

Then, both algorithms are second-order:

**Proposition A.1.** *Under the above assumptions, when $h_{max}$ is sufficiently small, we have for both Algorithms 1 and 2, $\|\boldsymbol{x}_{t_M} - \tilde{\boldsymbol{x}}_{t_M}\| = O(h_{max}^2)$.*

### A.2.1  CONVERGENCE OF ALGORITHM 1

The convergence proof of this algorithm is similar to that in DPM-Solver-2 (Lu et al., 2022). We give it in this section for completeness.

First, Taylor's expansion gives

$$
\boldsymbol{x}_{s_i} = \frac{\alpha_{s_i}}{\alpha_{t_{i-1}}}\boldsymbol{x}_{t_{i-1}} - \alpha_{s_i}(e^{-r_i h_i} - 1)\boldsymbol{x}_\theta(\boldsymbol{x}_{t_{i-1}}, t_{i-1}) + O(h_i^2),
$$

$$
\boldsymbol{x}_{t_i} = \frac{\alpha_{t_i}}{\alpha_{t_{i-1}}}\boldsymbol{x}_{t_{i-1}} - \alpha_{t_i}(e^{-h_i} - 1)\boldsymbol{x}_\theta(\boldsymbol{x}_{t_{i-1}}, t_{i-1}) - \alpha_{t_i}\left(-e^{-h_i} - h_i + 1\right)\boldsymbol{x}_\theta^{(1)}(\boldsymbol{x}_{t_{i-1}}, t_{i-1}) + O(h_i^3).
$$

Let $\Delta_i := \|\tilde{\boldsymbol{x}}_{t_i} - \boldsymbol{x}_{t_i}\|$, then $\|\boldsymbol{u}_i - \boldsymbol{x}_{s_i}\| \le C\Delta_{i-1} + CLh_i\Delta_{i-1} + Ch_i^2$. Note that

$$
\left\| \boldsymbol{x}_\theta^{(1)}(\boldsymbol{x}_{t_{i-1}}, t_{i-1}) - \frac{1}{r_i h_i}\left(\boldsymbol{x}_\theta(\boldsymbol{x}_{s_i}, s_i) - \boldsymbol{x}_\theta(\boldsymbol{x}_{t_{i-1}}, t_{i-1})\right)\right\| \le Ch_i.
$$

Since $r_i$ is bounded away from zero, and $e^{-h_i} = 1 - h_i + h_i^2/2 + O(h_i^3)$, we know

$$
\begin{aligned}
&\left\|(-e^{-h_i} - h_i + 1)\boldsymbol{x}_\theta^{(1)}(\boldsymbol{x}_{t_{i-1}}, t_{i-1}) - \frac{e^{-h_i} - 1}{2r_i}\left(\boldsymbol{x}_\theta(\boldsymbol{u}_i, s_i) - \boldsymbol{x}_\theta(\tilde{\boldsymbol{x}}_{t_{i-1}}, t_{i-1})\right)\right\| \\
&\le CLh_i(\Delta_{i-1} + \|\boldsymbol{u}_i - \boldsymbol{x}_{s_i}\|) + Ch_i^3 \\
&\quad + \frac{1}{r_i}\left|\frac{e^{-h_i} - 1}{2} - \frac{-e^{-h_i} - h_i + 1}{h_i}\right| \|\boldsymbol{x}_\theta(\boldsymbol{x}_{s_i}, s_i) - \boldsymbol{x}_\theta(\boldsymbol{x}_{t_{i-1}}, t_{i-1})\| \\
&\le CLh_i(\Delta_{i-1} + \|\boldsymbol{u}_i - \boldsymbol{x}_{s_i}\|) + Ch_i^3 + Ch_i^2\|\boldsymbol{x}_\theta(\boldsymbol{x}_{s_i}, s_i) - \boldsymbol{x}_\theta(\boldsymbol{x}_{t_{i-1}}, t_{i-1})\| \\
&\le CLh_i\Delta_{i-1} + C(L + M_i)h_i^3,
\end{aligned}
$$

where $M_i = 1 + \sup_{t_{i-1} \leq t \leq t_i} \|x_\theta^{(1)}(x_t, t)\|$. Then, $\Delta_i$ could be estimated as follows.

$$\Delta_i \leq \frac{\alpha_{t_i}}{\alpha_{t_{i-1}}} \Delta_{i-1} + \tilde{C}h_i(\Delta_{i-1} + h_i^2).$$

Thus, $\Delta_i = O(h_{max}^2)$ as long as $h_{max}$ is sufficiently small.

### A.3 CONVERGENCE OF ALGORITHM 2

Following the same line of argument of the convergence proof of Algorithm 1, we can prove the convergence of Algorithm 2. Let $\Delta_i := \|\tilde{x}_{t_i} - x_{t_i}\|$. Taylor's expansion yields

$$\left\| x_{t_i} - \left( \frac{\alpha_{t_i}}{\alpha_{t_{i-1}}} x_{t_{i-1}} - \alpha_{t_i}(e^{-h_i} - 1)x_\theta(x_{t_{i-1}}, t_{i-1}) - \alpha_{t_i} \left( -e^{-h_i} - h_i + 1 \right) x_\theta^{(1)}(x_{t_{i-1}}, t_{i-1}) \right) \right\| \leq Ch_i^3,$$

where $C$ is a constant depends on $x_\theta^{(2)}$. Also note that

$$\left\| x_\theta^{(1)}(x_{t_{i-1}}, t_{i-1}) - \frac{1}{h_{i-1}} \left( x_\theta(x_{t_{i-1}}, t_{i-1}) - x_\theta(x_{t_{i-2}}, t_{i-2}) \right) \right\| \leq Ch_i,$$

Since $r_i$ is bounded away from zero, and $e^{-h_i} = 1 - h_i + h_i^2/2 + O(h_i^3)$, we know

$$\left\| (-e^{-h_i} - h_i + 1)x_\theta^{(1)}(x_{t_{i-1}}, t_{i-1}) - \frac{e^{-h_i} - 1}{2r_i} \left( x_\theta(\tilde{x}_{t_{i-1}}, t_{i-1}) - x_\theta(\tilde{x}_{t_{i-2}}, t_{i-2}) \right) \right\|$$

$$\leq CLh_i(\Delta_{i-1} + \Delta_{i-2}) + Ch_i^3 + \frac{1}{r_i} \left| \frac{e^{-h_i} - 1}{2} - \frac{-e^{-h_i} - h_i + 1}{h_i} \right| \left\| x_\theta(x_{t_{i-1}}, t_{i-1}) - x_\theta(x_{t_{i-2}}, t_{i-2}) \right\|$$

$$\leq CLh_i(\Delta_{i-1} + \Delta_{i-2}) + Ch_i^3 + Ch_i^2 \left\| x_\theta(x_{t_{i-1}}, t_{i-1}) - x_\theta(x_{t_{i-2}}, t_{i-2}) \right\|$$

$$\leq CLh_i(\Delta_{i-1} + \Delta_{i-2}) + CM_i h_i^3,$$

where $M_i = 1 + \sup_{t_{i-1} \leq t \leq t_i} \|x_\theta^{(1)}(x_t, t)\|$. Then, $\Delta_i$ could be estimated as follows.

$$\Delta_i \leq \frac{\alpha_{t_i}}{\alpha_{t_{i-1}}} \Delta_{i-1} + \alpha_{t_i}(1 - e^{-h_i})L\Delta_{i-1} + \alpha_{t_i} \left( CM_i h_i^3 + CLh_i(\Delta_{i-1} + \Delta_{i-2}) \right) + Ch_i^3$$

$$\leq \frac{\alpha_{t_i}}{\alpha_{t_{i-1}}} \Delta_{i-1} + \tilde{C}h_i(\Delta_{i-1} + \Delta_{i-2} + h_i^2).$$

Thus, $\Delta_i = O(h_{max}^2)$ as long as $h_{max}$ is sufficiently small and $\Delta_0 + \Delta_1 = O(h_{max}^2)$, which can be verified by the Taylor's expansion.

## B COMPARISON BETWEEN DPM-SOLVER AND DPM-SOLVER++

In this section, we convert DPM-Solver++(2S) to the formulation w.r.t. the noise prediction model and compare it with the second-order DPM-Solver (Lu et al., 2022).

At each step, the second-order DPM-Solver (DPM-Solver-2 (Lu et al., 2022)) has the following updating:

$$u_i = \frac{\alpha_{s_i}}{\alpha_{t_{i-1}}} \tilde{x}_{t_{i-1}} - \sigma_{s_i}(e^{r_i h_i} - 1)\epsilon_\theta(\tilde{x}_{t_{i-1}}, t_{i-1})$$

$$\tilde{x}_{t_i} = \frac{\alpha_{t_i}}{\alpha_{t_{i-1}}} \tilde{x}_{t_{i-1}} - \sigma_{t_i}(e^{h_i} - 1)\epsilon_\theta(\tilde{x}_{t_{i-1}}, t_{i-1}) \tag{10}$$

$$- \frac{\sigma_{t_i}}{2r_i}(e^{h_i} - 1)\left( \epsilon_\theta(u_i, s_i) - \epsilon_\theta(\tilde{x}_{t_{i-1}}, t_{i-1}) \right)$$

while DPM-Solver++(2S) has the following updating:

$$u_i = \frac{\sigma_{s_i}}{\sigma_{t_{i-1}}} \tilde{x}_{t_{i-1}} - \alpha_{s_i}(e^{-r_i h_i} - 1)x_\theta(\tilde{x}_{t_{i-1}}, t_{i-1})$$

$$\tilde{x}_{t_i} = \frac{\sigma_{t_i}}{\sigma_{t_{i-1}}} \tilde{x}_{t_{i-1}} - \alpha_{t_i}(e^{-h_i} - 1)\left( (1 - \frac{1}{2r_i})x_\theta(\tilde{x}_{t_{i-1}}, t_{i-1}) + \frac{1}{2r_i}x_\theta(u_i, s_i) \right) \tag{11}$$

Because

$$\boldsymbol{x}_\theta(\boldsymbol{x}, t) = \frac{\boldsymbol{x} - \sigma_t \boldsymbol{\epsilon}_\theta(\boldsymbol{x}, t)}{\alpha_t} = \frac{1}{\alpha_t} \boldsymbol{x} - e^{-\lambda_t} \boldsymbol{\epsilon}_\theta(\boldsymbol{x}, t)$$

we can rewrite DPM-Solver++(2S) w.r.t. the noise prediction model (see Appendix B.1 for details):

$$\boldsymbol{u}_i = \frac{\alpha_{s_i}}{\alpha_{t_{i-1}}} \tilde{\boldsymbol{x}}_{t_{i-1}} - \sigma_{s_i}(e^{r_i h_i} - 1)\boldsymbol{\epsilon}_\theta(\tilde{\boldsymbol{x}}_{t_{i-1}}, t_{i-1})$$

$$\tilde{\boldsymbol{x}}_{t_i} = \frac{\alpha_{t_i}}{\alpha_{t_{i-1}}} \tilde{\boldsymbol{x}}_{t_{i-1}} - \sigma_{t_i}(e^{h_i} - 1)\boldsymbol{\epsilon}_\theta(\tilde{\boldsymbol{x}}_{t_{i-1}}, t_{i-1}) \tag{12}$$

$$- \frac{\sigma_{t_i}}{2r_i}(e^{h_i} - 1)\underbrace{e^{-r_i h_i}}_{<1}\big(\boldsymbol{\epsilon}_\theta(\boldsymbol{u}_i, s_i) - \boldsymbol{\epsilon}_\theta(\tilde{\boldsymbol{x}}_{t_{i-1}}, t_{i-1})\big)$$

Comparing with Eq. 10, we can find that the only difference between DPM-Solver-2 and DPM-Solver++(2S) is that DPM-Solver++(2S) has an additional coefficient $e^{-r_i h_i} < 1$ at the second term (which is corresponding to approximating the first-order total derivative $\boldsymbol{\epsilon}_\theta^{(1)}$). Specifically, we have

$$\boldsymbol{\epsilon}_\theta(\boldsymbol{u}_i, s_i) - \boldsymbol{\epsilon}_\theta(\tilde{\boldsymbol{x}}_{t_{i-1}}, t_{i-1}) = \boldsymbol{\epsilon}_\theta^{(1)}(\tilde{\boldsymbol{x}}_{t_{i-1}}, t_{i-1}) + \mathcal{O}(h_i)$$

As DPM-Solver++(2S) multiplies a smaller coefficient into the $\mathcal{O}(h_i)$ error term, the constant before the high-order error term of DPM-Solver++(2S) is smaller than that of DPM-Solver-2. As they both are equivalent to a second-order discretization of the diffusion ODE, a smaller constant before the error term can result in a smaller discretization error and reducing the numerical instabilities (especially for large guidance scales). Therefore, using the data prediction model is a key for stabilizing the sampling, and DPM-Solver++(2S) is more stable than DPM-Solver-2.

## B.1 DETAILED DERIVATION

We can rewrite DPM-Solver++(2S) by:

$$\boldsymbol{u}_i = \frac{\sigma_{s_i}}{\sigma_{t_{i-1}}} \tilde{\boldsymbol{x}}_{t_{i-1}} - \alpha_{s_i}(e^{-r_i h_i} - 1)\boldsymbol{x}_\theta(\tilde{\boldsymbol{x}}_{t_{i-1}}, t_{i-1})$$

$$= \frac{\sigma_{s_i}}{\sigma_{t_{i-1}}} \tilde{\boldsymbol{x}}_{t_{i-1}} - \frac{\alpha_{s_i}}{\alpha_{t_{i-1}}}(e^{-\lambda_{s_i} + \lambda_{t_{i-1}}} - 1)\tilde{\boldsymbol{x}}_{t_{i-1}} + \alpha_{s_i}(e^{-\lambda_{s_i}} - e^{-\lambda_{t_{i-1}}})\boldsymbol{\epsilon}_\theta(\tilde{\boldsymbol{x}}_{t_{i-1}}, t_{i-1})$$

$$= \frac{\alpha_{s_i}}{\alpha_{t_{i-1}}} \tilde{\boldsymbol{x}}_{t_{i-1}} - \sigma_{s_i}(e^{r_i h_i} - 1)\boldsymbol{\epsilon}_\theta(\tilde{\boldsymbol{x}}_{t_{i-1}}, t_{i-1})$$

and

$$\tilde{\boldsymbol{x}}_{t_i} = \frac{\sigma_{t_i}}{\sigma_{t_{i-1}}} \tilde{\boldsymbol{x}}_{t_{i-1}} - \alpha_{t_i}(e^{-h_i} - 1)\left((1 - \frac{1}{2r_i})\boldsymbol{x}_\theta(\tilde{\boldsymbol{x}}_{t_{i-1}}, t_{i-1}) + \frac{1}{2r_i}\boldsymbol{x}_\theta(\boldsymbol{u}_i, s_i)\right)$$

$$= \frac{\alpha_{t_i}}{\alpha_{t_{i-1}}} \tilde{\boldsymbol{x}}_{t_{i-1}} - \sigma_{t_i}(e^{h_i} - 1)\boldsymbol{\epsilon}_\theta(\tilde{\boldsymbol{x}}_{t_{i-1}}, t_{i-1}) - \frac{\alpha_{t_i}}{2r_i}(e^{-h_i} - 1)(\boldsymbol{x}_\theta(\boldsymbol{u}_i, s_i) - \boldsymbol{x}_\theta(\tilde{\boldsymbol{x}}_{t_{i-1}}, t_{i-1}))$$

$$= \frac{\alpha_{t_i}}{\alpha_{t_{i-1}}} \tilde{\boldsymbol{x}}_{t_{i-1}} - \sigma_{t_i}(e^{h_i} - 1)\boldsymbol{\epsilon}_\theta(\tilde{\boldsymbol{x}}_{t_{i-1}}, t_{i-1}) + \frac{\sigma_{t_i}}{2r_i}(e^{h_i} - 1)e^{\lambda_{t_{i-1}}}(\boldsymbol{x}_\theta(\boldsymbol{u}_i, s_i) - \boldsymbol{x}_\theta(\tilde{\boldsymbol{x}}_{t_{i-1}}, t_{i-1}))$$

and

$$e^{\lambda_{t_{i-1}}}\big(\boldsymbol{x}_\theta(\boldsymbol{u}_i, s_i) - \boldsymbol{x}_\theta(\tilde{\boldsymbol{x}}_{t_{i-1}}, t_{i-1})\big)$$

$$= e^{\lambda_{t_{i-1}}}\left(\frac{1}{\alpha_{s_i}}\boldsymbol{u}_i - \frac{1}{\alpha_{t_{i-1}}}\tilde{\boldsymbol{x}}_{t_{i-1}} - e^{-\lambda_{s_i}}\boldsymbol{\epsilon}_\theta(\boldsymbol{u}_i, s_i) + e^{-\lambda_{t_{i-1}}}\boldsymbol{\epsilon}_\theta(\tilde{\boldsymbol{x}}_{t_{i-1}}, t_{i-1})\right)$$

$$= e^{\lambda_{t_{i-1}}}\left(-e^{-\lambda_{s_i}}(e^{\lambda_{s_i} - \lambda_{t_{i-1}}} - 1)\boldsymbol{\epsilon}_\theta(\tilde{\boldsymbol{x}}_{t_{i-1}}, t_{i-1}) - e^{-\lambda_{s_i}}\boldsymbol{\epsilon}_\theta(\boldsymbol{u}_i, s_i) + e^{-\lambda_{t_{i-1}}}\boldsymbol{\epsilon}_\theta(\tilde{\boldsymbol{x}}_{t_{i-1}}, t_{i-1})\right)$$

$$= e^{\lambda_{t_{i-1}}}\big(e^{-\lambda_{s_i}}\boldsymbol{\epsilon}_\theta(\tilde{\boldsymbol{x}}_{t_{i-1}}, t_{i-1}) - e^{-\lambda_{s_i}}\boldsymbol{\epsilon}_\theta(\boldsymbol{u}_i, s_i)\big)$$

$$= e^{-r_i h_i}\big(\boldsymbol{\epsilon}_\theta(\tilde{\boldsymbol{x}}_{t_{i-1}}, t_{i-1}) - \boldsymbol{\epsilon}_\theta(\boldsymbol{u}_i, s_i)\big)$$

so we have

$$\tilde{\boldsymbol{x}}_{t_i} = \frac{\alpha_{t_i}}{\alpha_{t_{i-1}}} \tilde{\boldsymbol{x}}_{t_{i-1}} - \sigma_{t_i}(e^{h_i} - 1)\boldsymbol{\epsilon}_\theta(\tilde{\boldsymbol{x}}_{t_{i-1}}, t_{i-1}) - \frac{\sigma_{t_i}}{2r_i}(e^{h_i} - 1)e^{-r_i h_i}\big(\boldsymbol{\epsilon}_\theta(\boldsymbol{u}_i, s_i) - \boldsymbol{\epsilon}_\theta(\tilde{\boldsymbol{x}}_{t_{i-1}}, t_{i-1})\big)$$

## C    IMPLEMENTATION DETAILS

### C.1    CONVERTING DISCRETE-TIME DPMS TO CONTINUOUS-TIME

Discrete-time DPMs (Ho et al., 2020) train the noise prediction model $\epsilon_\theta$ at $N$ fixed time steps $\{t_n\}_{n=1}^N$ and the noise prediction model is parameterized by $\tilde{\epsilon}_\theta(x_n, \frac{1000n}{N})$ for $n = 1, \ldots, N$, where each $x_n$ is corresponding to the value at time $t_n$. In practice, these discrete-time DPMs usually choose uniform time steps between $[0, T]$, thus $t_n = \frac{nT}{N}$, for $n = 1, \ldots, N$. The smallest time is $\frac{T}{N}$.

Moreover, for the widely-used DDPM (Ho et al., 2020), we usually choose a sequence $\{\beta_n\}_{n=1}^N$ which is defined by either linear schedule (Ho et al., 2020) or cosine schedule (Nichol & Dhariwal, 2021). After obtained the $\beta_n$ sequence, the noise schedule $\alpha_n$ is defined by

$$\alpha_n = \prod_{i=1}^n (1 - \beta_n), \tag{13}$$

where each $\alpha_n$ is corresponding to the continuous-time $t_n = \frac{nT}{N}$, i.e. $\alpha_{t_n} = \alpha_n$. To generalize the discrete $\alpha_n$ to the continuous version, we use a linear interpolation for the function $\log \alpha_n$. Specifically, for each $t \in [t_n, t_{n+1}]$, we define

$$\log \alpha_t := \log \alpha_n + \frac{\log \alpha_{n+1} - \log \alpha_n}{t_{n+1} - t_n}(t - t_n). \tag{14}$$

Therefore, we can obtain a continuous-time noise schedule $\alpha_t$ defined for all $t \in [\frac{T}{N}, T]$, and the std $\sigma_t = \sqrt{1 - \alpha_t^2}$ and the logSNR $\lambda_t = \log \alpha_t - \log \sigma_t$. Moreover, the logSNR $\lambda_t$ is strictly decreasing for $t$, thus the change-of-variable for $\lambda$ is still valid.

In practice, we usually have $T = 1$ and $N = 1000$, thus the smallest time is $10^{-3}$. Therefore, we solve the diffusion ODEs from time $t = 1$ to time $t = 10^{-3}$ to get our final sample. Such sampling can reduce the first-order discrete-time DDIM solver when using a uniform time step.

### C.2    ABLATING TIME STEPS

Previous DEIS only tuned on low-resolutional data CIFAR-10, which may be not suitable for high-resolutional data such as ImageNet 256x256 and large guidance scales for guided sampling. For a fair comparison with the baseline samplers, we firstly do ablation study for the time steps with the pretrained DPMs (Dhariwal & Nichol, 2021) on ImageNet 256x256 and vary the classifier guidance scale. In our experiments, we tune the time step schedule according to their power function choices. Specifically, let $t_M = 10^{-3}$ and $t_0 = 1$, the time steps $\{t_i\}_{i=0}^M$ satisfies

$$t_i = \left(\frac{M - i}{M} t_0^{\frac{1}{\kappa}} + \frac{i}{M} t_M^{\frac{1}{\kappa}}\right)^\kappa,$$

where $\kappa$ is a hyperparameter. Following Zhang & Chen (2022), we search $\kappa$ in $1, 2, 3$ by DEIS, and the results are shown in Table 2. We find that for all guidance scales, the best setting is $\kappa = 1$, i.e. the uniform $t$ for time steps. We further compare uniform $t$ and uniform $\lambda$ and find that the uniform $t$ time step schedule is still the best choice. Therefore, in all of our experiments, we use the uniform $t$ for evaluations.

### C.3    EXPERIMENT SETTINGS

We use uniform time step schedule for all experiments. Particularly, as DPM-Solver (Lu et al., 2022) is designed for uniform $\lambda$ (the intermediate time steps are a half of the step size w.r.t. $\lambda$), we also convert the intermediate time steps to ensure all the time steps are uniform $t$. We find that such conversion can improve the sample quality of both the singlestep DPM-Solver the singlestep DPM-Solver++.

We run NFE in 10, 15, 20, 25 for the high-order solvers and additional 50, 100, 250 for DDIM. For all experiments, we solver diffusion ODEs from $t = 1$ to $t = 10^{-3}$ with the interpolation of noise schedule detailed in Appendix C.1. For DEIS, we use the "t-AB-$k$" methods for $k = 1, 2, 3$, which is the fastest method in their original paper, and we name them as DEIS-$k$, respectively.

For the sampled image in Fig. 5, we use the prompt "A beautiful castle beside a waterfall in the woods, by Josef Thoma, matte painting, trending on artstation HQ".

Table 2: Sample quality measured by FID ↓ on ImageNet 256×256 (discrete-time model (Dhariwal & Nichol, 2021)), varying the methods between DDIM (Song et al., 2021a) and different types of DEIS (Zhang & Chen, 2022). The number of function evaluations (NFE) is fixed by 10.

| Method \ Guidance scale | 8.0 | 7.0 | 6.0 | 5.0 | 4.0 | 3.0 | 2.0 | 1.0 | 0.0 |
|---|---|---|---|---|---|---|---|---|---|
| DDIM | **13.04** | **12.38** | **11.81** | 11.55 | 11.62 | 11.95 | 13.01 | 16.35 | 29.33 |
| DEIS-2, $\kappa = 1$ | 19.12 | 14.83 | 12.39 | **10.94** | **10.13** | **9.76** | **9.74** | 11.01 | 20.34 |
| DEIS-2, $\kappa = 2$ | 33.37 | 24.66 | 18.03 | 13.57 | 11.16 | 10.54 | 10.88 | 13.67 | 26.26 |
| DEIS-2, $\kappa = 3$ | 55.69 | 44.01 | 33.04 | 24.50 | 18.66 | 16.35 | 16.87 | 21.91 | 38.41 |
| DEIS-3, $\kappa = 1$ | 66.81 | 48.71 | 33.89 | 22.56 | 15.84 | 11.96 | 10.18 | **10.19** | **18.70** |
| DEIS-3, $\kappa = 2$ | 34.51 | 25.42 | 18.52 | 13.68 | 11.20 | 10.46 | 10.75 | 13.36 | 25.59 |
| DEIS-3, $\kappa = 3$ | 56.49 | 44.51 | 33.34 | 24.68 | 18.72 | 16.38 | 16.79 | 21.76 | 38.02 |

# D    EXPERIMENT DETAILS

We list all the detailed experimental results in this section.

Table 3: Sample quality measured by FID ↓ on ImageNet 256×256 (discrete-time model (Dhariwal & Nichol, 2021)), varying the number of function evaluations (NFE).

| Guidance Scale | Thresholding | Sampling Method \ NFE | 10 | 15 | 20 | 25 | 50 | 100 | 250 |
|---|---|---|---|---|---|---|---|---|---|
| 8.0 | No | DDIM (Song et al., 2021a) | 13.04 | 11.27 | 10.21 | 9.87 | 9.82 | 9.52 | 9.37 |
| | | PNDM (Liu et al., 2022b) | 99.80 | 37.59 | 15.50 | 11.54 | \ | \ | \ |
| | | DPM-Solver-2 (Lu et al., 2022) | 114.62 | 44.05 | 20.33 | 9.84 | \ | \ | \ |
| | | DPM-Solver-3 (Lu et al., 2022) | 164.74 | 91.59 | 64.11 | 29.40 | \ | \ | \ |
| | | DEIS-1 (Zhang & Chen, 2022) | 15.20 | 10.86 | 10.26 | 10.01 | \ | \ | \ |
| | | DEIS-2 (Zhang & Chen, 2022) | 19.12 | 11.37 | 10.08 | 9.75 | \ | \ | \ |
| | | DEIS-3 (Zhang & Chen, 2022) | 66.86 | 24.48 | 12.98 | 10.87 | \ | \ | \ |
| | | DPM-Solver++(S) (ours) | **12.20** | 9.85 | 9.19 | 9.32 | \ | \ | \ |
| | | DPM-Solver++(M) (ours) | 14.44 | **9.46** | **9.10** | **9.11** | \ | \ | \ |
| | Yes | DDIM (Song et al., 2021a) | 10.58 | 9.53 | 9.12 | 8.94 | 8.58 | 8.49 | 8.48 |
| | | DPM-Solver++(S) (ours) | **9.26** | 8.93 | **8.40** | 8.63 | \ | \ | \ |
| | | DPM-Solver++(M) (ours) | 9.56 | **8.64** | 8.50 | **8.39** | \ | \ | \ |
| 4.0 | No | DDIM (Song et al., 2021a) | 11.62 | 9.67 | 8.96 | 8.58 | 8.22 | 8.06 | 7.99 |
| | | PNDM (Liu et al., 2022b) | 22.71 | 10.03 | 8.69 | 8.47 | \ | \ | \ |
| | | DPM-Solver-2 (Lu et al., 2022) | 37.68 | 9.42 | 8.22 | 8.08 | \ | \ | \ |
| | | DPM-Solver-3 (Lu et al., 2022) | 74.97 | 15.65 | 9.99 | 8.15 | \ | \ | \ |
| | | DEIS-1 (Zhang & Chen, 2022) | 10.55 | 9.47 | 8.88 | 8.65 | \ | \ | \ |
| | | DEIS-2 (Zhang & Chen, 2022) | 10.13 | 9.09 | 8.68 | 8.45 | \ | \ | \ |
| | | DEIS-3 (Zhang & Chen, 2022) | 15.84 | 9.25 | 8.63 | 8.43 | \ | \ | \ |
| | | DPM-Solver++(S) (ours) | 9.08 | 8.51 | **8.00** | 8.07 | \ | \ | \ |
| | | DPM-Solver++(M) (ours) | **8.98** | **8.26** | 8.06 | **8.06** | \ | \ | \ |
| | Yes | DDIM (Song et al., 2021a) | 10.45 | 8.95 | 8.51 | 8.25 | 7.91 | 7.82 | 7.87 |
| | | DPM-Solver++(S) (ours) | 8.94 | 8.26 | **7.95** | **7.87** | \ | \ | \ |
| | | DPM-Solver++(M) (ours) | **8.91** | **8.21** | 7.99 | 7.96 | \ | \ | \ |
| 2.0 | No | DDIM (Song et al., 2021a) | 13.01 | 9.60 | 9.02 | 8.45 | 7.72 | 7.60 | 7.44 |
| | | PNDM (Liu et al., 2022b) | 11.58 | 8.48 | 8.17 | 7.97 | \ | \ | \ |
| | | DPM-Solver-2 (Lu et al., 2022) | 14.12 | 8.20 | 8.59 | 7.48 | \ | \ | \ |
| | | DPM-Solver-3 (Lu et al., 2022) | 21.06 | 8.57 | 8.19 | 7.85 | \ | \ | \ |
| | | DEIS-1 (Zhang & Chen, 2022) | 10.40 | 9.11 | 8.52 | 8.21 | \ | \ | \ |
| | | DEIS-2 (Zhang & Chen, 2022) | 9.74 | 8.80 | 8.28 | 8.06 | \ | \ | \ |
| | | DEIS-3 (Zhang & Chen, 2022) | 10.18 | 8.63 | 8.20 | 7.98 | \ | \ | \ |
| | | DPM-Solver++(S) (ours) | **9.18** | 8.17 | **7.77** | 7.56 | \ | \ | \ |
| | | DPM-Solver++(M) (ours) | 9.19 | 8.47 | 8.17 | 8.07 | \ | \ | \ |
| | Yes | DDIM (Song et al., 2021a) | 11.19 | 9.20 | 8.42 | 8.05 | 7.65 | 7.59 | 7.63 |
| | | DPM-Solver++(S) (ours) | **9.23** | 8.18 | **7.81** | **7.60** | \ | \ | \ |
| | | DPM-Solver++(M) (ours) | 9.28 | 8.56 | 8.28 | 8.18 | \ | \ | \ |

Table 4: Sample quality measured by MSE ↓ on COCO2014 validation set (discrete-time latent model (Rombach et al., 2022)), varying the number of function evaluations (NFE). Guidance scale is 7.5, which is the recommended setting for stable-diffusion (Rombach et al., 2022).

| Guidance Scale | Thresholding | Sampling Method \ NFE | 10 | 15 | 20 | 25 | 50 | 100 | 250 |
|---|---|---|---|---|---|---|---|---|---|
| | | DDIM (Song et al., 2021a) | 0.59 | 0.42 | 0.48 | 0.45 | 0.34 | 0.23 | 0.12 |
| | | PNDM (Liu et al., 2022b) | 0.66 | 0.43 | 0.50 | 0.46 | 0.32 | \ | \ |
| | | DPM-Solver-2 (Lu et al., 2022) | 0.66 | 0.47 | 0.40 | 0.34 | 0.20 | \ | \ |
| 7.5 | No | DPM-Solver-3 (Lu et al., 2022) | 0.59 | 0.48 | 0.43 | 0.37 | 0.23 | \ | \ |
| | | DEIS-1 (Zhang & Chen, 2022) | **0.47** | **0.39** | **0.34** | **0.29** | 0.16 | \ | \ |
| | | DEIS-2 (Zhang & Chen, 2022) | 0.48 | 0.40 | **0.34** | **0.29** | **0.15** | \ | \ |
| | | DEIS-3 (Zhang & Chen, 2022) | 0.57 | 0.45 | 0.38 | 0.34 | 0.19 | \ | \ |
| | | DPM-Solver++(S) (**ours**) | 0.48 | 0.41 | 0.36 | 0.32 | 0.19 | \ | \ |
| | | DPM-Solver++(M) (**ours**) | 0.49 | 0.40 | **0.34** | **0.29** | 0.16 | \ | \ |
| | | DDIM (Song et al., 2021a) | **0.83** | 0.78 | 0.71 | 0.67 | \ | \ | \ |
| | | PNDM (Liu et al., 2022b) | 0.99 | 0.87 | 0.79 | 0.75 | \ | \ | \ |
| | | DPM-Solver-2 (Lu et al., 2022) | 1.13 | 1.08 | 0.96 | 0.86 | \ | \ | \ |
| 15.0 | No | DEIS-1 (Zhang & Chen, 2022) | 0.84 | **0.72** | **0.64** | **0.58** | \ | \ | \ |
| | | DEIS-2 (Zhang & Chen, 2022) | 0.87 | 0.76 | 0.68 | 0.63 | \ | \ | \ |
| | | DEIS-3 (Zhang & Chen, 2022) | 1.06 | 0.88 | 0.78 | 0.73 | \ | \ | \ |
| | | DPM-Solver++(S) (**ours**) | 0.88 | 0.75 | 0.68 | 0.61 | \ | \ | \ |
| | | DPM-Solver++(M) (**ours**) | 0.84 | **0.72** | **0.64** | **0.58** | \ | \ | \ |

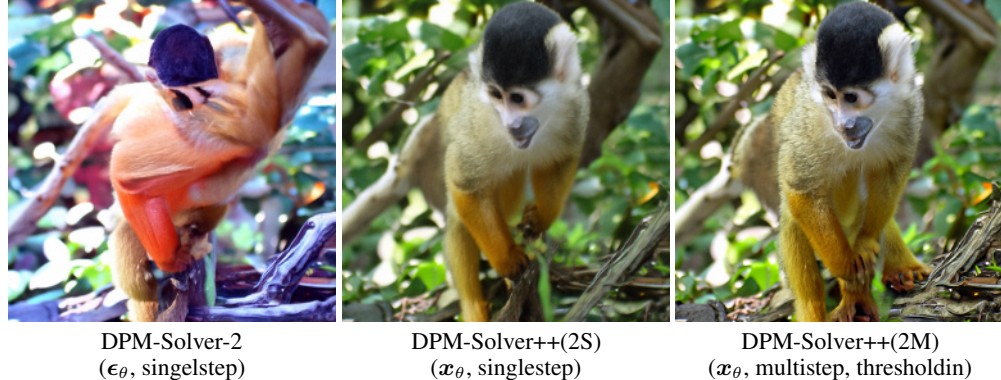

DPM-Solver-2
($\epsilon_\theta$, singelstep)

DPM-Solver++(2S)
($x_\theta$, singlestep)

DPM-Solver++(2M)
($x_\theta$, multistep, thresholdin)

Figure 4: Samples of different sampling methods for DPMs on ImageNet 256x256 with guidance scale 8.0.

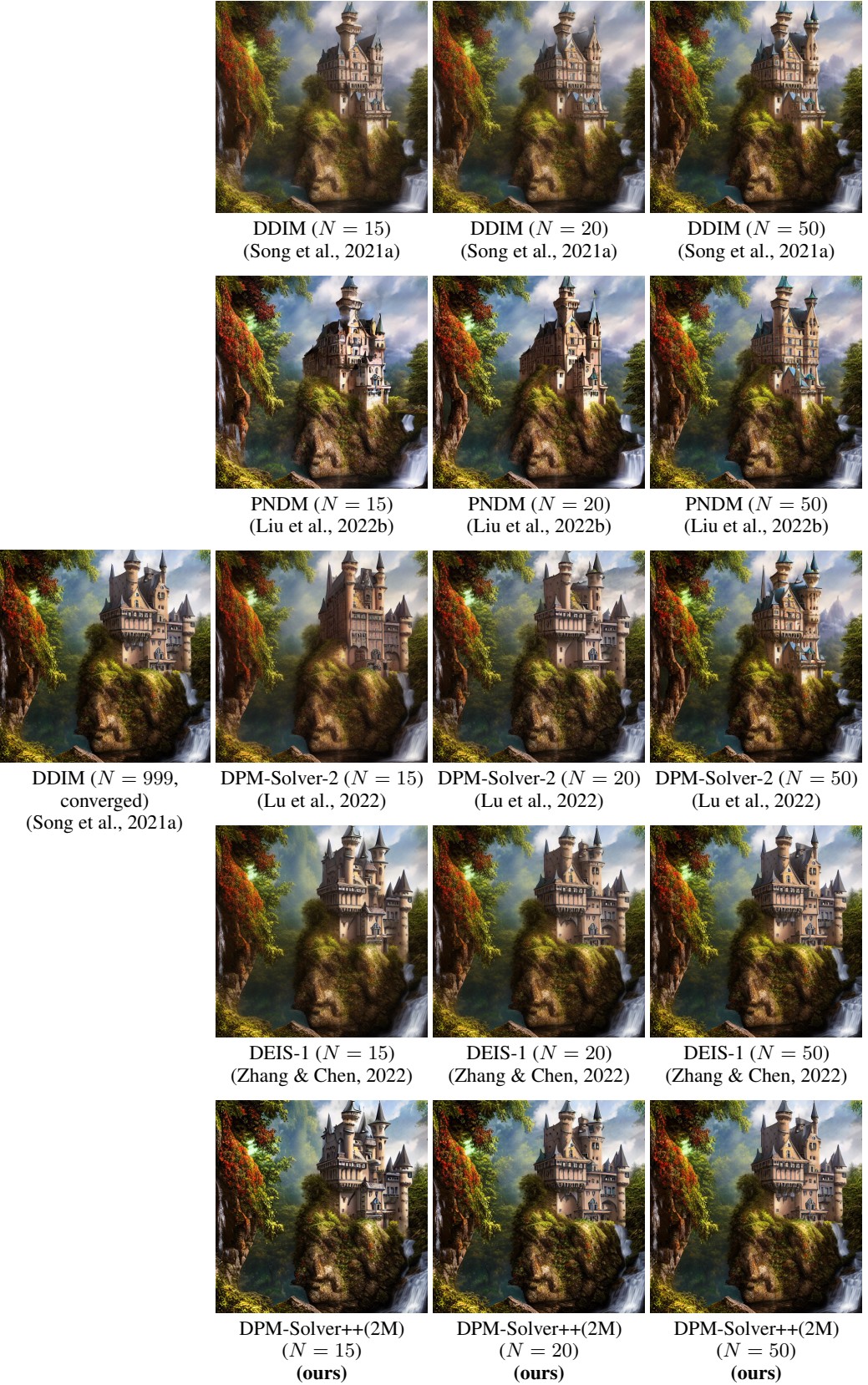

Figure 5: Samples using the pre-trained latent-space DPMs (Stable-Diffusion (Rombach et al., 2022)) with a classifier-free guidance scale 7.5 (the default setting), varying different samplers and different number of function evaluations $N$.