# OpenReview forum: "DPM-Solver++: Fast Solver for Guided Sampling of Diffusion Probabilistic Models"
_ICLR.cc/2023/Conference — Submitted to ICLR 2023_

### Official Review · Reviewer_8HFn · 2022-10-24

**Confidence:** 4
**Correctness:** 3
**Technical Novelty And Significance:** 2
**Empirical Novelty And Significance:** 3
**Recommendation:** 6

**Clarity, Quality, Novelty And Reproducibility:**

**Clarity:** The paper is overall well-written and easy to follow. However, I am missing some details about how exactly the thresholding is implemented in DPM-Solver++ and also about why other methods cannot use it (see discussion above). This is currently difficult to follow. Also more discussions on x- vs. epsilon-prediction would make the paper stronger (also see above).

**Quality:** Overall, the paper is of good quality for the most part. It's written well, has thorough experiments, and the work is put nicely into context.

**Novelty:** As discussed in *Weaknesses, point 1* above, the methodological novelty of the paper is rather incremental (see discussion there).

**Reproducibility:** Overall, the paper seems sufficiently easily reproducible. All solver details and hyperparameters are provided.

**Strength And Weaknesses:**

**Strengths:**

- Investigating fast higher-order solvers for diffusion model sampling specifically for the guidance scenario is a good idea and practically very important, due to the wide usage of guidance in the diffusion model literature.
- The shown experimental results are strong and the new sampling approach works well in almost all settings. It usually outperforms all existing methods.

**Weaknesses:**

- The technical novelty of the solver is small. It is a direct follow-up of DPM-Solver, using the same framework. Changing from epsilon- to x-prediction is simple and applying thresholding is also trivial. The multistepping approach has also been used previously in the diffusion modeling literature.

- Why exactly does x-prediction work better than epsilon-prediction (in isolation, without thresholding)? This should be properly discussed.

- The paper points out that other higher-order baselines such as DEIS and DPM-Solver cannot use thresholding in their higher-order forms. I am not sure this is entirely correct. Unless I am misunderstanding something, I would think that also DEIS and DPM-Solver can always take their epsilon prediction, convert into x-prediction via $x_\theta = (x_t - \sigma_t \epsilon_\theta)/\alpha_t$, apply the threshold, and convert back to epsilon prediction (and then just use the corresponding frameworks). I think the paper should analyze these details much more thoroughly and discuss in more detail. I would recommend the authors to lay out in detail how exactly thresholding is applied in DPM-Solver++ in the higher-order scheme and why this shouldn't be possible in the baselines --- because it looks to me like it was possible. Moreover, if the other methods can use thresholding, after all, the experimental baseline comparisons would have to be adapted accordingly.

- The paper writes that it searches over step size schedules and then uses the same schedule in all experiments for all solvers. Maybe I am confusing something, but I am not sure this is an ideal approach. I think using the same step size schedule in all experiments could potentially be unfair to methods that may have been designed with other schedules in mind. I think for every baseline the best possible step size schedule should be chosen. It sounds as if only DEIS and DPM-Solver++ are considered when finding step size schedules. DPM-Solver and PNDM could potentially have other optimal choices.

Minor question: In Figure 1, does DPM-Solver++ use thresholding or not? It would be interesting to add the synthesized image for DPM-Solver++ both with and without thresholding.

**Summary Of The Paper:**

This paper tackles fast sampling from diffusion models when using guidance. Guided sampling has become a crucial ingredient in conditional diffusion models to achieve high synthesis quality. However, most existing solvers for accelerated synthesis are not developed specifically with guidance in mind. In fact, the paper shows that some previous higher-order methods for fast sampling perform poorly in the guidance setting. The paper then proposes the DPM-Solver++, a higher-order solver that is based on an exponential integrator framework, similar to previous work. The approach is essentially a direct follow-up of DPM-Solver, tailored to the guidance setting. Compared to previous works, there are two crucial differences: (a) the new solver works with the x-prediction diffusion model parametrization, in contrast to previous works using epsilon-prediction. (b) This enables an easy application of thresholding algorithms to prevent samples from going out of bounds when performing the iterative synthesis process with large guidance weights. Moreover, the paper develops its novel solver both with a single-step framework and a multi-stepping framework.

Experimentally, the novel DPM-Solver++ compares favourably to a variety of baselines on various fast synthesis tasks. The solver is applied on conditional pixel space diffusion models as well as on Stable Diffusion, a latent text-to-image diffusion model.

**Summary Of The Review:**

In summary, the paper studies an important problem, fast sampling from conditional diffusion models using guidance with higher-order solvers. However, I think the claims made around other higher-order methods not being able to apply thresholding are either incorrect, or should be discussed in more detail to avoid confusion. Furthermore, the conceptual novelty of the paper is rather small. Experimentally, the results appear strong, but arguably guidance and thresholding should be applied for all baselines, if possible (see discussion on this above).

In conclusion, I believe the paper is not quite ready yet for publication in its current form. I am willing to increase my paper rating, if my concerns can be addressed and potential confusions resolved.

---

> ### Author Response · Authors · 2022-11-18
> **Thank you for the valuable feedback!**
>
> We thank reviewer 8HFn for the interest and acknowledgment of our strong empirical contributions and the insightful questions. Below we respond to the questions. We would highly appreciate it if the reviewer agree with our response and consider to raise the score. Thank you so much!
>
> **Q1. The novelty is small**
>
> We agree with the reviewer's comments that the main techniques for designing DPM-Solver++ are similar to DPM-Solver. However, we would like to emphasize that this work is not a incremental extension of DPM-Solver, but **a dedicated solution for accelerating the guided sampling** by diffusion models. The technical novelty in this work has two main aspects:
>
> 1. To the best of our knowledge, **this is the first work to reveal the fact that high-order solvers have instability issues** in guided sampling with large guidance scales.
>
>    We would like to emphasize that such finding is also important and useful for the community, because it is highly counterintuitive from previous findings in fast solvers (e.g., both DPM-Solver-3 and DEIS-3 works well in accelerating the CIFAR-10 unconditional sampling, but suffer from terribly numerical issues in guided sampling). Therefore, **despite the previous success for accelerating unconditional sampling, designing a fast high-order solver for guided sampling is still an open and difficult problem**.
>
> 2. We further provide **a theoretical analysis to demonstrate why using data prediction model can reduce the instability issue**.
>
>    In the revision paper, we also add a theoretical comparison between DPM-Solver-2 and DPM-Solver++(2S) in Appendix B to explain why simply using data prediction model can reduce the instability issue. Such analysis is also our techique novelty.
>
> In summary, in this work, we dig deeply in designing high-order fast solvers for guided sampling with large guidance scales, and successfully accelerate guided sampling by the proposed DPM-Solver++.
>
> Moreover, we would like to argue that **the empirical success is also important to the whole community**, because most of the downstream applications of diffusion models are guided sampling (e.g. text-to-image, image editing, etc.) and the sampling speed is crucial for the downstream tasks. We believe the proposed DPM-Solver++ is easy to use in all the downstream tasks and can greatly promote the application of diffusion models.
>
> **Q2. Why exactly does x-prediction work better than epsilon-prediction (even without thresholding)?**
>
> We've added a theoretical comparison between DPM-Solver-2 and DPM-Solver++(2S) in Appendix B to explain why simply using data prediction model can reduce the instability issue. In short, DPM-Solver++(2S) has a **smaller constant before the high-order error term**. Therefore, although both are second-order solvers, DPM-Solver++(2S) has smaller discretization error than DPM-Solver-2.
>
> **Q3. Other solvers can also apply the thresholding method**
>
> We sincerely thank the reviewer for pointing out our misunderstanding. We've corrected our statements in the revision and weakened the claims about the thresholding. We'd like to emphasize that the thresholding method is not our key contribution and is just a engineering trick to further improve the sample quality in pixel-space. Even without thresholding, DPM-Solver++ can achieve the SOTA accleration performance in both pixel-space and latent-space diffusion models (Table 3&4), and thus our main conclusion stays the same.
>
> However, we believe that even all applying with thresholding, DPM-Solver++ can still outperform the baselines (because of our theoretical analysis that DPM-Solver++ has a smaller discretization error). Due to the time limitation of the rebuttal, we will add the detailed results in our final version.
>
> **Q4. Step size schedule searching for baselines**
>
> Ideally, we should search all possible step size schedules for all baselines and for all steps. However, searching a generally optimal time step schedule is an open problem in the literature and we do not focus on solving such problem.  In this work, we only search for the step size schedules used in previous works (DEIS, DPM-Solver, PNDM) and find that:
>
> 1. The degree of the influence of the time step schedule highly depends on the dataset, but is almost invariant to the total steps (i.e. NFEs).
> 2. For unconditional sampling on low-resolutional CIFAR-10, DEIS and PNDM can achieve a better sample quality with their proposed $\kappa$ spacing (as provided in DEIS paper), and DPM-Solver can achieve a better sample quality with their proposed logSNR spacing (as provided in DPM-Solver paper); However, for high-resolutional guided sampling (especially with large guidance scales), both the $\kappa$ spacing and the logSNR spacing works much worse **(highly visible to the human eyes)** than the uniform time steps. We also provide a quantative result in Table 2 for completeness.
>
> Therefore, we indeed use the best setting for the baselines: uniform time steps.

---

> > ### Author Response · Authors · 2022-11-18
> > **Part 2**
> >
> >
> > **Q5. Figures of DPM-Solver++ without thresholding**
> >
> > We provide the result in Fig.4. to demonstrate that DPM-Solver++(2S) without thresholding is much better than previous baselines (without thresholding).

---

> > > ### Comment · Reviewer_8HFn · 2022-11-22
> > > **Thank you for the response**
> > >
> > > I would like to thank the authors for their reply and answering my questions. I appreciate that the authors improved the paper and fixed the error with regards to the possibility of thresholding in other algorithms. I also like the theoretical analysis why x-prediction alone works better than epsilon prediction here. Overall, I think the technical novelty of the paper is still relatively small, but I am now more convinced that the method could be practically useful. Therefore, I raised my score by one point and am now carefully leaning towards suggesting acceptance.

---

> > > > ### Author Response · Authors · 2022-11-23
> > > > **Thank you for raising the score!**
> > > >
> > > > Thank you so much for raising the score. We believe the practical success of DPM-Solver++ will have a high impact on the community on many downstream guided sampling tasks.

---

### Official Review · Reviewer_s8Si · 2022-10-24

**Confidence:** 3
**Correctness:** 4
**Technical Novelty And Significance:** 2
**Empirical Novelty And Significance:** 2
**Recommendation:** 5

**Clarity, Quality, Novelty And Reproducibility:**

The idea is clearly explained and well supported by the experiments. The combination of applying data prediction model and thresholding also seems to be novel in this context.

**Strength And Weaknesses:**

Strength
- experimental results show significant improvement over DDIM for ImageNet in 15-20 steps

Weakness
- The idea is not principled and this reviewer is confused to find a principled way to think about the generality of the idea for handling instability for distributions beyond the reported experiments. It is not quite clear why switching to a data prediction model avoids the instability. Clipping is also a common method for such scenarios and not surprising.


Questions and comments
- About latent diffusions and experiments with stable diffusions, is there instability observed as in the pixel space? The latent space seems to be smoother and as a result the higher order derivatives of the score could be more bounded. Can the instability still be a serious issue there?


**Summary Of The Paper:**

this submission deals with accelerating guided sampling of DPMs. The challenge however is that high-order samplers based on noise prediction become unstable for large guidance steps, and they suffer from test-train mismatch. To address these challenges, this work proposes an ODE solver based on the data prediction model that approximates the integrator of the score with Taylor approximation and calculates the first-order and second-order derivatives. In addition, a clipping is used to keep the solution within the range. Experiments with both pixel and latent space DPMs indicate that it can generate high fidelity images in 15-20 steps.


**Summary Of The Review:**

This paper addresses an interesting and timely topic. The contribution however seems limited. It simply switches to the model prediction method and uses the common clipping to avoid instability. None of this seems to be a surprise, and thus the significance and generality of the idea is not clear.

---

> ### Author Response · Authors · 2022-11-18
> **Thank you for the valuable feedback!**
>
> We thank reviewer s8Si for the interest and acknowledgment of our empirical contributions and the insightful questions. Below we respond to the questions. We would highly appreciate it if the reviewer agree with our response and consider to raise the score. Thank you so much!
>
> **Q1. Is this a principled way for handling instability for distributions beyond the reported experiments?**
>
> Sorry for the lack of clarity. We'd like to emphasize that the proposed DPM-Solver++ is a principled way for stabilizing the sampling by diffusion models. The reason is:
>
> 1. DPM-Solver++ is dedicated for diffusion ODEs and **analytically computes all the terms that are independent of the neural network**. Specifically, as shown in Eq.(9), DPM-Solver++ analytically computes the linear term and the coefficients before the derivatives of the model. Thus, the only discretization error of DPM-Solver++ is due to the approximation of the derivatives of the data prediction model.
>
> 2. Comparing with DPM-Solver, the proposed DPM-Solver++ has a **smaller constant before the high-order error term**. Therefore, DPM-Solver++ is **both theoretically and empirically** more stable than DPM-Solver.
>
>    We've added a theoretical comparison between DPM-Solver-2 and DPM-Solver++(2S) in Appendix B to explain why simply using data prediction model can reduce the instability issue. In short, DPM-Solver++(2S) has a smaller constant before the high-order error term. Therefore, although both are second-order solvers, DPM-Solver++(2S) has smaller discretization error than DPM-Solver-2.
>
> In summary, theoretically speaking, the proposed DPM-Solver++ reduces the discretization error as much as possible and the discretization error of DPM-Solver++ is even smaller than DPM-Solver. Therefore, it is a principled way for accelerating sampling by diffusion models, especially better than baselines in guided sampling with large guidance scales.
>
> **Q2. Why switching to a data prediction model avoids the instability?**
>
> We've added a theoretical comparison between DPM-Solver-2 and DPM-Solver++(2S) in Appendix B to explain why simply using data prediction model can reduce the instability issue. In short, DPM-Solver++(2S) has a **smaller constant before the high-order error term**. Therefore, although both are second-order solvers, DPM-Solver++(2S) has smaller discretization error than DPM-Solver-2.
>
> Note that such analysis is independent of the thresholding trick. This can explain why DPM-Solver++(2S) is empirically better than DPM-Solver-2 even without thresholding.
>
> **Q3. Is the instability observed in latent-space diffusion as in the pixel space?**
>
> Yes, the instability issues occur for both latent-space diffusions and pixel-space diffusions with large enough guidance scales. However, the latent-space diffusions can tolerate larger guidance scales than the pixel-space diffusions. We agree with the reviewer's explaination that the latent space is smoother and as a result the higher order derivatives of the score could be more bounded.
>
> We've added the results for stable-diffusion with 15.0 guidance scale in Table 4. The conclusion is the same as the pixel-space:
>
> 1. High-order solvers suffer from instability issues.
> 2. DPM-Solver++(2S) is much better than DPM-Solver-2 (both without thresholding).
> 3. DPM-Solver++(2M) achieves the SOTA acceleration result.
>
> **Q4. Can the instability still be a serious issue in latent-space diffusions?**
>
> As the default setting for stable-diffusion is 7.5, the instability issue is not as that bad as in pixel-space diffusions. However, even for the 7.5 guidance scale, the proposed DPM-Solver++ can still achieve a SOTA acceleration performance.
>
> Moreover, we would like to argue that the instability issue highly depends on the guidance scale, and the guidance scale highly depends on the specific task. In the experiment of this work, we only consider the widely-used text-to-image task with stable-diffusion for the latent-space cases. However, there may exist a downstream task which needs a large enough guidance scale and in that case the instability issue will still be a seriour issue for the latent-space diffusion.

---

> > ### Author Response · Authors · 2022-11-29
> > **Sincerely looking forward to the further discussions**
> >
> > Dear reviewer,
> >
> > We are wondering if our response and revision have resolved your concerns. If our response has addressed your concerns, we would highly appreciate it if you could re-evaluate our work and consider raising the score.
> >
> > If you have any additional questions or suggestions, we would be happy to have further discussions.
> >
> > Best regards,
> >
> > The Authors

---

### Official Review · Reviewer_zAj6 · 2022-10-25

**Confidence:** 4
**Correctness:** 1
**Technical Novelty And Significance:** 2
**Empirical Novelty And Significance:** 2
**Recommendation:** 5

**Clarity, Quality, Novelty And Reproducibility:**

Clarity: ok

Quality: questionable

Novelty: questionable

Reproducibility: ok

**Strength And Weaknesses:**

Strength:

* Authors propose data prediction model from noise prediction model based on parameterization and its corresponding ODEs.

Weakness:

* The algorithm is quite similar to previous works, DEIS and DPM-Solver. The improvement is incremental.

* It would be nice if the author could add a comparison between Algorithm1 and DPM-Solver-2 in the language of the same model and highlight the difference. Because the data prediction model is induced by $x_\theta = (x_t - \sigma_t \epsilon) / \alpha_t$, Algorithm1 should be able to be rewritten as noise prediction model too.

* Table 1 is biased. First, both DEIS and DPM-Solver can be combined with thresholding easily. Notice that $x_\theta = (x_t - \sigma_t \epsilon) / \alpha_t$, we can infer corresponding "thresholded" $\epsilon$ from "thresholded" $x_\theta$ based on $\epsilon = (x_t - \alpha_t x_\theta) / \sigma_t $.
Second, DEIS can be shown as a high-order method and enjoy high-order convergence. If the reviewer understands it correctly, the analytical form can be obtained by exponential transformation introduced in DEIS~(see sec 4 in DEIS).

* Therefore, most experiments are not comprehensive and need to show the performance of PNDM/DPM/DEIS with thresholding.

**Summary Of The Paper:**

The work proposes a fast ODE solver for the guided sampling of diffusion probabilistic models with parameterization and thresholding. However, experiments and comparisons are not fair.

**Summary Of The Review:**

The work proposes a fast ODE solver for the guided sampling of diffusion probabilistic models with parameterization and thresholding. However, experiments and comparisons are not fair. I would like to increase the score if authors can address the questions I raised.

---

> ### Author Response · Authors · 2022-11-18
> **Thank you for the valuable feedback!**
>
>
> We thank reviewer zAj6 for the interest and acknowledgment of our empirical contributions and the insightful questions. Below we respond to the questions. We would highly appreciate it if the reviewer agree with our response and consider to raise the score. Thank you so much!
>
> **Q1. The improvement is incremental?**
>
> We agree with the reviewer's comments that the main techniques for designing DPM-Solver++ are similar to DPM-Solver and DEIS. However, we would like to emphasize that this work is not a incremental extension of DPM-Solver, but **a dedicated solution for accelerating the guided sampling** by diffusion models. The technical novelty in this work has two main aspects:
>
> 1. To the best of our knowledge, **this is the first work to reveal the fact that high-order solvers have instability issues** in guided sampling with large guidance scales.
>
>    We would like to emphasize that such finding is also important and useful for the community, because it is highly counterintuitive from previous findings in fast solvers (e.g., both DPM-Solver-3 and DEIS-3 works well in accelerating the CIFAR-10 unconditional sampling, but suffer from terribly numerical issues in guided sampling). Therefore, **despite the previous success for accelerating unconditional sampling, designing a fast high-order solver for guided sampling is still an open and difficult problem**.
>
> 2. We further provide **a theoretical analysis to demonstrate why using data prediction model can reduce the instability issue**.
>
>    In the revision paper, we also add a theoretical comparison between DPM-Solver-2 and DPM-Solver++(2S) in Appendix B to explain why simply using data prediction model can reduce the instability issue. Such analysis is also our techique novelty.
>
> In summary, in this work, we dig deeply in designing high-order fast solvers for guided sampling with large guidance scales, and successfully accelerate guided sampling by the proposed DPM-Solver++.
>
> Moreover, we would like to argue that **the empirical success is also important to the whole community**, because most of the downstream applications of diffusion models are guided sampling (e.g. text-to-image, image editing, etc.) and the sampling speed is crucial for the downstream tasks. We believe the proposed DPM-Solver++ is easy to use in all the downstream tasks and can greatly promote the application of diffusion models.
>
> **Q2. Comparison between DPM-Solver++2S and DPM-Solver-2**
>
> We've added a theoretical comparison between DPM-Solver-2 and DPM-Solver++(2S) in Appendix B to explain why simply using data prediction model can reduce the instability issue. In short, DPM-Solver++(2S) has a **smaller constant before the high-order error term**. Therefore, although both are second-order solvers, DPM-Solver++(2S) has smaller discretization error than DPM-Solver-2.
>
> Note that such analysis is independent of the thresholding trick. This can explain why DPM-Solver++(2S) is empirically better than DPM-Solver-2 even without thresholding.
>
> **Q3. Correcting the Statements of DEIS in Table 1**
>
> We sincerely thank the reviewer for pointing out our misunderstanding. We've corrected our statements in the revision.
>
> **Q4. Other solvers can also apply the thresholding method**
>
> We sincerely thank the reviewer for pointing out our misunderstanding. We've corrected our statements in the revision and weakened the claims about the thresholding. We'd like to emphasize that the thresholding method is not our key contribution and is just a engineering trick to further improve the sample quality in pixel-space. Even without thresholding, DPM-Solver++ can achieve the SOTA accleration performance in both pixel-space and latent-space diffusion models (Table 3&4), and thus our main conclusion stays the same.
>
> However, we believe that even all applying with thresholding, DPM-Solver++ can still outperform the baselines (because of our theoretical analysis that DPM-Solver++ has a smaller discretization error). Due to the time limitation of the rebuttal, we will add the detailed results in our final version.

---

> > ### Author Response · Authors · 2022-11-29
> > **Sincerely looking forward to the further discussions**
> >
> > Dear reviewer,
> >
> > We are wondering if our response and revision have resolved your concerns. If our response has addressed your concerns, we would highly appreciate it if you could re-evaluate our work and consider raising the score.
> >
> > If you have any additional questions or suggestions, we would be happy to have further discussions.
> >
> > Best regards,
> >
> > The Authors

---

> > > ### Comment · Reviewer_zAj6 · 2022-12-09
> > > **reply**
> > >
> > > Thanks for the authors’ response. I decide to raise the score because of the empirical performance of DPM-solver++ and the author’s response. However, some claims and arguments need further explanations.
> > >
> > > DPM-Solver++ has a smaller coefficient for second order term
> > > I appreciate authors make a clear comparison between DPM-Solver and DPM-Solver++. The only difference lies in the coefficient for the second-order term.
> > >
> > > I have some difficulties understanding why it works. Authors claim the coefficient can help reduce high-order error, however, the coefficient also contributes the approximated $\epsilon_\theta^{(1)}$. In another word, if we multiply the second-order term by zero, we get zero for the high-order error term but it also destroys acceleration brought by  $\epsilon_\theta^{(1)}$. Therefore, I do not think the author’s explanation can justify the advantage of DPM-Solver++ over DPM-Solver.
> > >
> > > Second, I notice the single-step DPM-Solver has a similar performance as the single-step DPM-Solver++ on unguided sampling in my preliminary experiments. Is it true that single-step DPM-Solver++ is better than single-step DPM-Solver when guided weights are large? If this is true, then it seems the authors’ response can not explain the observation.
> > >
> > > In the last, some empirical comparison against $\epsilon$-model + ‘thresholding` will make the conclusion more convincing.

---

### Official Review · Reviewer_a1rL · 2022-11-02

**Confidence:** 4
**Clarity, Quality, Novelty And Reproducibility:** See above
**Correctness:** 4
**Technical Novelty And Significance:** 2
**Empirical Novelty And Significance:** 3
**Recommendation:** 5

**Strength And Weaknesses:**

[Pros.]
- Good practical extension of DPM-Solver for guided sampling
- The presentation is clear and easy to follow

[Cons.]
- The technical novelty behind this work is rather incremental. The main techniques used in this paper are directly borrowed from its predecessor [A] (e.g, semi-linear diffusion ODE solver, exponential integrators) without significant deltas. The innovations introduced are the data prediction formulation of the diffusion ODE and the multistep solver, which is good but not surprising. It seems the key to stabilizing sampling is owing to the dynamic thresholding method, which also comes from another paper [B]. As a result, it's hard to evaluate the technical novelty of this paper, the proposed method looks sound but is more of an "engineering" trick
- Could you please explain why the data prediction parameterization yields better results then noise prediction one even without using thresholding?

[A] DPM-Solver: A Fast ODE Solver for Diffusion Probabilistic Model Sampling in Around 10 Steps, NeurIPS 2022
[B] Photorealistic text-to-image diffusion models with deep language understanding, arxiv 2022




**Summary Of The Paper:**

This paper focuses on a practical aspect of diffusion-based generative models, i.e., training-free fast guided sampling of DDPM. It is a direct extension of the recently proposed fast unconditional sampler namely DPM-Solver.  The authors identify the unstable sampling trajectory of DPM-Solver for guided sampling, and present several tricks to resolve this issue. These include 1) reformulating the diffusion ODE with data prediction parameterization in lieu of noise prediction; 2) deriving the corresponding ODE higher-order solver leveraging the semi-linear structure of diffusion ODE and exponential integrators; 3) combining the solver with the thresholding trick. The numerical results show these techniques successfully resolve the convergence issue of the DPM-Solver for guided sampling, in both pixel space and latent space settings.

**Summary Of The Review:**

IMO, the ideas presented in this manuscript serve a nice contribution as a journal extension of its predecessor but are not appropriate to be published as another top-tier conference paper.

---

> ### Author Response · Authors · 2022-11-18
> **Thank you for the valuable feedback!**
>
> We thank reviewer a1rL for the interest and acknowledgment of our empirical contributions and the insightful questions. Below we respond to the questions. We would highly appreciate it if the reviewer agree with our response and consider to raise the score. Thank you so much!
>
> **Q1. The novelty is rather incremental?**
>
> We agree with the reviewer's comments that the main techniques for designing DPM-Solver++ are similar to DPM-Solver. However, we would like to emphasize that this work is not a incremental extension of DPM-Solver, but **a dedicated solution for accelerating the guided sampling** by diffusion models. The technical novelty in this work has two main aspects:
>
> 1. To the best of our knowledge, **this is the first work to reveal the fact that high-order solvers have instability issues** in guided sampling with large guidance scales.
>
>    We would like to emphasize that such finding is also important and useful for the community, because it is highly counterintuitive from previous findings in fast solvers (e.g., both DPM-Solver-3 and DEIS-3 works well in accelerating the CIFAR-10 unconditional sampling, but suffer from terribly numerical issues in guided sampling). Therefore, **despite the previous success for accelerating unconditional sampling, designing a fast high-order solver for guided sampling is still an open and difficult problem**.
>
> 2. We further provide **a theoretical analysis to demonstrate why using data prediction model can reduce the instability issue**.
>
>    In the revision paper, we also add a theoretical comparison between DPM-Solver-2 and DPM-Solver++(2S) in Appendix B to explain why simply using data prediction model can reduce the instability issue. Such analysis is also our techique novelty.
>
> In summary, in this work, we dig deeply in designing high-order fast solvers for guided sampling with large guidance scales, and successfully accelerate guided sampling by the proposed DPM-Solver++.
>
> Moreover, we would like to argue that **the empirical success is also important to the whole community**, because most of the downstream applications of diffusion models are guided sampling (e.g. text-to-image, image editing, etc.) and the sampling speed is crucial for the downstream tasks. We believe the proposed DPM-Solver++ is easy to use in all the downstream tasks and can greatly promote the application of diffusion models.
>
> **Q2. The key to stabilizing sampling is owing to the dynamic thresholding method?**
>
> Sorry for the misleading in our paper. We've added a theoretical analysis in Appendix B to show that using data prediction model is another key for stabilizing sampling because it has a smaller constant before the higher-order error term.
>
> **Q3. Why the data prediction parameterization yields better results even without thresholding?**
>
> We've added a theoretical comparison between DPM-Solver-2 and DPM-Solver++(2S) in Appendix B to explain why simply using data prediction model can reduce the instability issue. In short, DPM-Solver++(2S) has a smaller constant before the high-order error term. Therefore, although both are second-order solvers, DPM-Solver++(2S) has smaller discretization error than DPM-Solver-2.

---

> > ### Author Response · Authors · 2022-11-29
> > **Sincerely looking forward to the further discussions**
> >
> > Dear reviewer,
> >
> > We are wondering if our response and revision have resolved your concerns. If our response has addressed your concerns, we would highly appreciate it if you could re-evaluate our work and consider raising the score.
> >
> > If you have any additional questions or suggestions, we would be happy to have further discussions.
> >
> > Best regards,
> >
> > The Authors

---

> > > ### Comment · Reviewer_a1rL · 2022-11-29
> > > **Thank you for the nice response**
> > >
> > > I want to acknowledge the authors did make a solid response to the concerns raised before, though I still preserve my opinion the novelty underlying is still somewhat limited.
> > >
> > > I appreciate that the authors could theoretically explain why data prediction parameterization yields better results, but the conclusion drawn seems too simple or "trivial" for me. What about heuristically setting a small constant before the higher-order error term? For practitioners who use DPM-Solver for guided sampling, it would be straightforward to realize the instability might come from errors induced in higher-order terms, such that heuristically introducing a small constant (<1) as an engineering trick is natural without complicating the algorithm.
> > >
> > > I'm willing to raise my score if the authors could give more evidence to highlight the importance of data prediction parameterization.

---

### Public Comment · ~Qinsheng_Zhang1 · 2022-11-18
**Thanks for comparing DEIS**

Thank you for comparing DEIS and conducting lots of experiments for DEIS in high-resolution datasets. However, I have some questions about the DPM-Solver++
1. It seems that DPM-Solver++ single step and DPM-Solver are equivalent when you choose the same timestamps scheduling in DPM-Solver, $s_i =t_\lambda((\lambda_i + \lambda_{i-1}) / 2)$ since line 6 only use the latest function evaluation similar to DPM-Solver. But experiments show there is a significant difference between the two algorithms.
2. It looks like DPM-Solver++ multiple step uses lower order warming start, line 4 is a first-order update step. In this case, will the convergence order will be bottlenecked by the warming start?

---

> ### Author Response · Authors · 2022-11-18
> **Thank you for the interest and valuable feedback!**
>
> We are happy that the first author of DEIS raised the question and discussion openly. We address the questions below and we believe this discussion can further improve the understanding of DPM-Solver++.
>
> **Q1. Comparison between DPM-Solver++2S and DPM-Solver-2**?
>
> In fact, they are different. We've added a theoretical comparison between DPM-Solver-2 and DPM-Solver++(2S) in Appendix B to explain why simply using data prediction model can reduce the instability issue. In short, DPM-Solver++(2S) has a smaller constant before the high-order error term. Therefore, although both are second-order solvers, DPM-Solver++(2S) has a smaller discretization error than DPM-Solver-2.
>
> **Q2. Convergence order with the warming start trick**?
>
> Thank you for pointing it out. I'm also a big fan of your great work DEIS, and the warming start trick in DPM-Solver++ actually follows DEIS-t-AB. However, I did not find any theoretical analysis of the convergence order with such trick in DEIS paper. Below I will introduce my own understanding of it. Feel free to discuss more because your opinion is also important to the community.
>
> 1. Empirically, the warming start trick can greatly improve the performance of the multistep solvers. Such conclusion stands for PNDM, DEIS, and DPM-Solver++2M.
>
> 2. Theoretically, we only apply the warming start trick at time T. Note that for the time near to T, the marginal distribution is almost a Gaussian distribution with std $\sigma_t$, and then the corresponding score function is almost $\nabla_{x}\log p_t(x_t)\approx -\frac{x_t}{\sigma_t}$. Thus, the noise prediction model is almost $\epsilon_\theta(x_t, t) \approx x_t$. In such cases, a first-order DDIM is enough for solving the exact solution (as discussed in [1], I'm also a big fan of your gDDIM work, so amazing!). Therefore, the warming start trick can approximate the true start point quite well, so the convergence order is not a serious problem and is still almost second order.
>
> [1] Zhang et al., gDDIM: Generalized denoising diffusion implicit models. https://arxiv.org/abs/2206.05564
>
> We sincerely appreciate the discussions with such an expert as you in the community. We believe the discussions can make our work clearer and promote the progress of the whole area. If you have any more questions or find some mistakes, please feel free to correct us. Thank you so much!

---

### Decision · Program_Chairs · 2023-01-20

**Decision:**

Reject

**Justification For Why Not Higher Score:**

Using existing techniques known to work for a problem to a problem with a different solver. It does not contain enough technical novelty to make the cut in my opinion.

**Justification For Why Not Lower Score:**

N/A

**Metareview: Summary, Strengths And Weaknesses:**

The paper extends the DPM-Solver for sampling with classifier-free guidance. Two major changes were made. First, it switches from the epsilon prediction in DPM-Solver to the x0 prediction. Second, it applies the dynamic thresholding trick proposed in the Imagen paper. Experiment results show it outperforms DPM-Solver.

The paper receives four reviews, with three reviewers rating the paper above the bar and one reviewer rating the paper below the bar. The negative reviewers are unhappy with the presented novelty and presentation. The positive reviewer thinks the presented algorithm is useful, and the presented results are convincing.

As there is no consensus, the AC must decide. After reading the paper, reviews, and rebuttal, the AC does not consider the paper a clear rejection. The AC agrees that the presented algorithm is useful and could be widely adopted. However, the AC does not feel the paper presents any surprising results. Both switching to the x0 prediction and using thresholding have been used in prior works. There is no surprise that they should help DPM-Solver improve the quality when applied in the classifier-free guidance setting.

**Summary Of Ac-Reviewer Meeting:**

N/A